# Condition-Aware Graph Flow Matching for Modeling the Distributions of Complex Fluid Systems

**Xiaochao Deng** [1 2]  **Jie Chen** [1]  **Xiaogang Deng** [1 2]

## Abstract

Accurately modeling the full distributions of possible states is crucial for understanding statistical properties and enabling reliable predictions in complex fluid systems. Recently, diffusion models and flow matching have shown promise in these tasks. However, they remain limited in uncovering the general principles of systems from multiple short trajectories across the condition space. In addition, they exhibit inferior adaptability to large irregular geometries, particularly in regions with sharp gradients. In this paper, we propose a condition-aware graph flow matching (CGFM) method that combines condition-aware flow matching with a hierarchical graph structure to learn the full distributions of fluid systems from incomplete training data. Specifically, CGFM constructs a flow enabling smooth interpolation across physical conditions and parameterizes the graph-conditioned vector field through HieraGraphNet. HieraGraphNet performs message passing across multilevel graphs to capture multi-scale dynamics and facilitate long-range information interactions in fluid systems. Moreover, we introduce a topology- and geometry-aware graph coarsening scheme that incorporates topological connectivity and local geometric density to construct reliable coarse graphs. We validate the effectiveness of CGFM on three canonical scenarios across both 2D and 3D dynamics, which demonstrate its superior performance compared with that of state-of-the-art baselines.

[1]School of Computer Science, Sichuan University, Chengdu, China [2]National Key Laboratory of Fundamental Algorithms and Models for Engineering Simulation, Sichuan University, Chengdu, China. Correspondence to: Jie Chen <chenjie2010@scu.edu.cn>.

*Proceedings of the 43rd International Conference on Machine Learning*, Seoul, South Korea. PMLR 306, 2026. Copyright 2026 by the author(s).

## 1. Introduction

Accurately modeling the underlying distribution (mechanisms ) of complex physical systems (e.g., fluid dynamics) is essential for understanding their statistical properties, and enabling robust decision-making in diverse applications, such as aerospace engineering for aerodynamic design (Lin et al., 2025; Zhao et al., 2024), climate science for weather forecasting (Yoon et al., 2024), and plasma physics for turbulence modeling (Castagna et al., 2024). Practitioners use numerical solvers to evolve system trajectories from the initial state until fully developed, converged flow states are reached. This process is often computationally expensive, making it impractical for real-world applications in which ensemble runs are needed to account for variability in physical conditions, e.g., optimization (Lin et al., 2025; Chen et al., 2025c) and control (Suárez et al., 2025).

A variety of deep neural network models have been developed to emulate numerical solvers in an autoregressive manner (Cao et al., 2023; Han et al., 2022; Lino et al., 2022; Pfaff et al., 2020). Notable instances include convolutional autoencoder-based models (Chen & Thuerey, 2024; Zhang & Chen, 2022) for rolling out grid-based physical fields, as well as graph neural network (GNN)-based models (Lei et al., 2025; Deng et al., 2025) for predicting unstructured flows over irregular domains. These models significantly improve the efficiency of simulating physical systems, which are considered surrogates for traditional numerical solvers. However, autoregressive models inevitably accumulate errors during iterative roll-outs, which often leads to instability in long-horizon forecasts (Lienen et al., 2024). For many chaotic systems (e.g., turbulent flows), their deterministic nature often fails to capture the intrinsic stochasticity, thus limiting their ability to generate diverse realizations (Du et al., 2024). More importantly, engineers typically focus on exploring the statistical properties and probability distributions of physical systems in statistical equilibrium (Pope, 2001) rather than specific solution trajectories. These observations motivate a new perspective: *can we directly model the distribution of equilibrium flow states from multiple short trajectories across different physical conditions, thereby enabling efficient sampling of diverse states and inexpensive computation of statistical properties?*

Recent advances in generative modeling, such as denoising diffusion probabilistic models (DDPMs) (Ho et al., 2020), score-based generative models (Song et al., 2021b), and flow matching (Lipman et al., 2023; Tong et al., 2023) have been successfully applied in diverse tasks, including generation of images (Hagemann et al., 2025; Li et al., 2025), speech (Lian et al., 2025; Wang et al., 2025a) and videos(Wang et al., 2025c; Jin et al., 2025), demonstrating remarkable capabilities in modeling the time-dependent evolution of probability distributions. These approaches still have significant limitations in modeling the dynamics of physical systems, as they neglect the intricate effects of conditioning variables on the resulting dynamics. Additionally, practical scientific measurements are often conducted at limited temporal trajectories and experimental settings because of the prohibitive costs of data acquisition. Flow matching provides a stable and sampling-efficient framework for scientific modeling tasks such as molecular generation (Zhou et al., 2025; Xu et al., 2025), protein design (Yan et al., 2025; Geffner et al., 2025), and sequence modeling (Chen et al.; Shukueian Tabrizi et al., 2025). However, two major challenges remain for extending flow matching to complex fluid systems: (i) *Modeling the underlying mechanisms from incomplete trajectories across multiple conditions.* Most existing generative models fail to jointly capture the shared dynamics of multiple related conditions, and explicitly reveal the full information of systems from incomplete trajectories. (ii) *Representing large-scale irregular systems.* The architectures of existing approaches predominantly operate on Cartesian grids, which limits their ability to flexibly represent irregular geometries that frequently arise in fluid dynamics and to adaptively allocate spatial resolution in these settings.

In this paper, we propose a condition-aware graph flow matching (CGFM) method that combines condition-aware flow matching with a hierarchical graph neural network to model the full distribution of the converged flow states of fluid systems from an ensemble of incomplete training trajectories. CGFM constructs a probability path enabling optimal interpolation between the source and equilibrium-state distribution and regresses it with a graph-conditioned vector field over graph-node features to jointly learn shared dynamics across trajectories. A hierarchical graph neural network (HieraGraphNet) is introduced to parameterize this vector field by operating on mesh-based graph structures, thereby flexibly representing irregular geometries and adaptively allocating spatial resolution in complex fluid systems. Moreover, we devise a topology- and geometry-aware graph coarsening scheme that considers both topological connectivity and local geometric density to effectively reduce dense connectivity in large graphs while preserving essential information. CGFM enables efficient sampling of physically plausible states from the equilibrium-state distribution by solving an ODE with only a few function evaluations, given the system's graph structure and physical parameters. These samples can then be used to derive arbitrary statistical properties with reasonable computational costs. This eliminates the need for long roll-outs of autoregressive models or ensemble simulations. We validate the effectiveness on both in-distribution and out-of-distribution datasets across three canonical 2D and 3D flow tasks. The results indicate that our model consistently outperforms state-of-the-art methods in terms of sample accuracy, distributional accuracy and flow statistics, while achieving superior interpolation and extrapolation capabilities in challenging settings.

In summary, our contributions are as follows:

- We present a CGFM method for modeling the equilibrium distribution of flow states, from which flow statistics can then be estimated by drawing multiple samples without long numerical simulations.

- We learn a graph-conditioned velocity field over graph-node features that captures shared patterns from multiple short trajectories such that its flow generates a desired probability path.

- A topology- and geometry-aware graph coarsening scheme is introduced to generate a coarse-to-fine hierarchy of unstructured mesh graphs. With this hierarchy, HieraGraphNet performs cross-level message passing to capture the multi-scale dynamic properties and facilitate the long-range interactions of physical contexts.

## 2. Related work

**Generative models**  Recent advances in probability path models, namely, flow matching (Lipman et al., 2023), eliminate the stochastic and iterative characteristics of diffusion models by learning a time-dependent vector field. Variants such as optimal transport paths (Tong et al., 2023), and rectified flows (Liu et al., 2023) further improve training stability and sample efficiency (Kornilov et al., 2024). Recent work has demonstrated the potential of flow matching for scientific and system modeling. Rohbeck et al. (2025) introduced multi-marginal flow matching, which extends flow matching to model dynamic systems across time and varying conditions, demonstrating its potential for the conditional generative modeling of trajectories. In the context of turbulence, Wang et al. (2025b) developed FourierFlow, a frequency-aware flow matching approach that explicitly incorporates spectral information to improve the fidelity of generated turbulent flows. Baldan et al. (2026) further proposed a unified framework that integrates flow matching with PDE-based constraints, enabling physics-informed generative modeling across a range of dynamical systems. These studies relied primarily on CNN- or spectral-based architectures for structured domains, thus leaving the potential

of flow matching on unstructured meshes and graph-based representations largely unexplored in large-scale fluid simulations.

**Graph-based models**   Graph structures provide a flexible representation for complex physical domains on unstructured meshes, and recent progress in graph structure learning and representation robustness (Lei et al., 2025; Lanteri et al., 2025; Chen et al., 2025a;b) has reinforced their importance. Leveraging graph representations, GNN-based simulators have evolved from stacked message passing (Allen et al., 2023; Pfaff et al., 2020) to attention-based (Sun et al., 2023; Han et al., 2022) and hierarchical U-Net-style architectures (Deng et al., 2025; Cao et al., 2023; Lino et al., 2022) for scalable and long-range information propagation. These advances have motivated the integration of GNNs with generative models for physical modeling. Valencia et al. (2025) proposed diffusion graph network (DGN), which couple DDPM with GNN to model the distribution of fluid simulations. DGN tend to produce high-frequency noise in generated samples, often suffer from long training times and thus require specialized acceleration strategies (Zhang & Chen, 2022; Song et al., 2021a). Recent works have extended flow matching to graph-based domains, including protein ensembles (Jing et al., 2024), molecular graph generation (Pandey et al., 2025), 3D molecular structure modeling (Zhou et al., 2025), social recommendation (Huang et al., 2025), and cellular morphology simulation (Zhang et al., 2025), which demonstrates its feasibility for graph representations.

## 3. Condition-aware Graph Flow Matching for System-State Distribution Modeling

### 3.1. Problem Definition

The dynamics of complex physical systems, such as unsteady fluid dynamics, are governed by partial differential equations (PDEs) of the form

$$\frac{\partial \mathbf{s}(\mathbf{x}, t)}{\partial t} = \mathcal{F}\big(\mathbf{s}(\mathbf{x}, t); \boldsymbol{\mu}\big), \quad (\mathbf{x}, t) \in \Omega \times [0, T], \quad (1)$$

where $\mathbf{s}(\mathbf{x}, t) \in \mathbb{R}^d$ denotes $d$-dimensional state quantities (e.g., velocity and pressure), $\mathcal{F}$ is a nonlinear operator parameterized by the physical parameter $\boldsymbol{\mu}$ (e.g., Reynolds number), and $\mathbf{x}$ denotes coordinates in the spatial domain $\Omega$. To numerically solve the PDEs, the spatial domain is discretized into an unstructured mesh $\mathcal{M} = (\mathcal{V}_{\mathcal{M}}, \mathcal{E}_{\mathcal{M}})$, where each mesh node $i \in \mathcal{V}_{\mathcal{M}}$ is located at $\mathbf{x}_i \in \Omega$. At time step $t$, the discrete system state $\mathbf{z}_t := \{\mathbf{s}_{i,t} \in \mathbb{R}^d : i \in \mathcal{V}_{\mathcal{M}}\} \in \mathbb{R}^{|\mathcal{V}_{\mathcal{M}}| \times d}$ is defined by the state quantities $\mathbf{s}_{i,t}$ at the mesh nodes. Let $\mathcal{C}$ denote the condition space, where each conditional variable $c \in \mathcal{C}$ defines the fluid systems, including the geometry, boundary conditions, and physical parameters. Given a physical condition $c$,

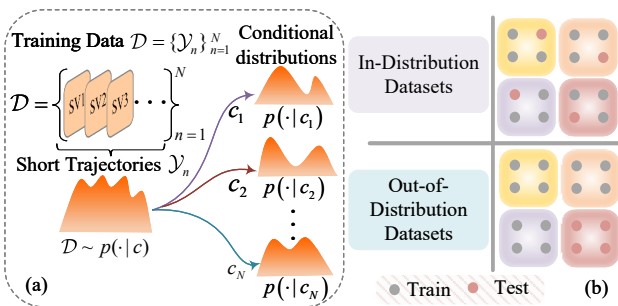

*Figure 1.* Dataset and generalization splits. (a) The training data $\mathcal{D}$ and their data distribution $p$. The dataset $\mathcal{D}$ contains $N$ short trajectories sampled from the converged states of the simulation systems. Each system state is described by the $d$-dimensional state quantities $\mathrm{SV}_1$, $\mathrm{SV}_2$, and $\mathrm{SV}_3$ (e.g., velocity and pressure). Conditioning variables $c$ induce conditional probability densities $p(\cdot|c)$. (b) In-distribution and out-of-distribution data splits used to validate the model's generation and generalization.

the numerical solver produces an infinite dynamical trajectory of states $\mathcal{Z} = \{\mathbf{z}_0, \mathbf{z}_1, \ldots, \mathbf{z}_t, \mathbf{z}_{t+1}, \ldots\}$ from an initial state $\mathbf{z}_0$. We assume that after a transient phase of length $t$, each system enters a *statistical equilibrium* regime $\mathcal{S} \subset \mathcal{Z}$, in which the distribution of states becomes independent of the initial state. In practice, long equilibrium state trajectories are rarely available due to prohibitive acquisition costs. Therefore, in our experiments, a short segment $\mathcal{Y} = \{\mathbf{z}_{t_0}, \mathbf{z}_{t_0+1}, \ldots, \mathbf{z}_{t_0+m-1}\}_{t_0 \geq t}$, $\mathcal{Y} \subset \mathcal{S}$, is randomly selected from the equilibrium regime of a trajectory $\mathcal{S}$. As shown in Figure 1, given a training dataset $\mathcal{D} = \{\mathcal{Y}_n\}_{n=1}^N$, we aim to learn a conditional probabilistic model $p_\theta(\cdot \mid c)$ that is capable of sampling of converged states $Y_t \in \mathcal{S}$ under unseen conditions.

### 3.2. Modeling Distribution over Graph-Node Features

We represent the simulation mesh of a fluid system $\mathcal{M}$ as a graph $\mathcal{G} = (\mathcal{V}, \mathcal{E})$. Mesh nodes $\mathcal{V}_{\mathcal{M}}$ become graph nodes $\mathcal{V}$, and mesh edges $\mathcal{E}_{\mathcal{M}}$ are represented as bidirectional mesh-edges $\mathcal{E}$ in the graph. The system state is encoded as node features on the mesh graph $\mathcal{G}$. The conditional information of the system is encoded as node and edge attributes on this graph $\mathcal{G}$, with node attributes $V_c = \{\mathbf{v}_i^c \mid i \in \mathcal{V}\}$ and edge attributes $E_c = \{\mathbf{e}_{ij}^c \mid (i, j) \in \mathcal{E}\}$. The system-specific attributes of the node and edge encodings are provided in Appendix B and Table 7. Our goal is to model the equilibrium-state distribution over graph-node features that can generate plausible samples $Y_t \in \mathbb{R}^{|\mathcal{V}_{\mathcal{M}}| \times d}$ conditioned on the system graph $\mathcal{G}$ and its node/edge attributes $(V_c, E_c)$. Accordingly, we construct a flow $\psi_\tau : \mathbb{R}^{|\mathcal{V}_{\mathcal{M}}| \times d} \to \mathbb{R}^{|\mathcal{V}_{\mathcal{M}}| \times d}$ over flow time $\tau \in [0, 1]$ on graph-node features $\mathbb{R}^{|\mathcal{V}_{\mathcal{M}}| \times d}$ that transports samples $Y_0$ from Gaussian distribution $p_0$ into target states $Y_1 := \psi_1(Y_0)$ such that $Y_1$ follows the desired equilibrium-state distribution $p_1(\cdot \mid \mathcal{G}, V_c, E_c)$. Specifically, we first construct a

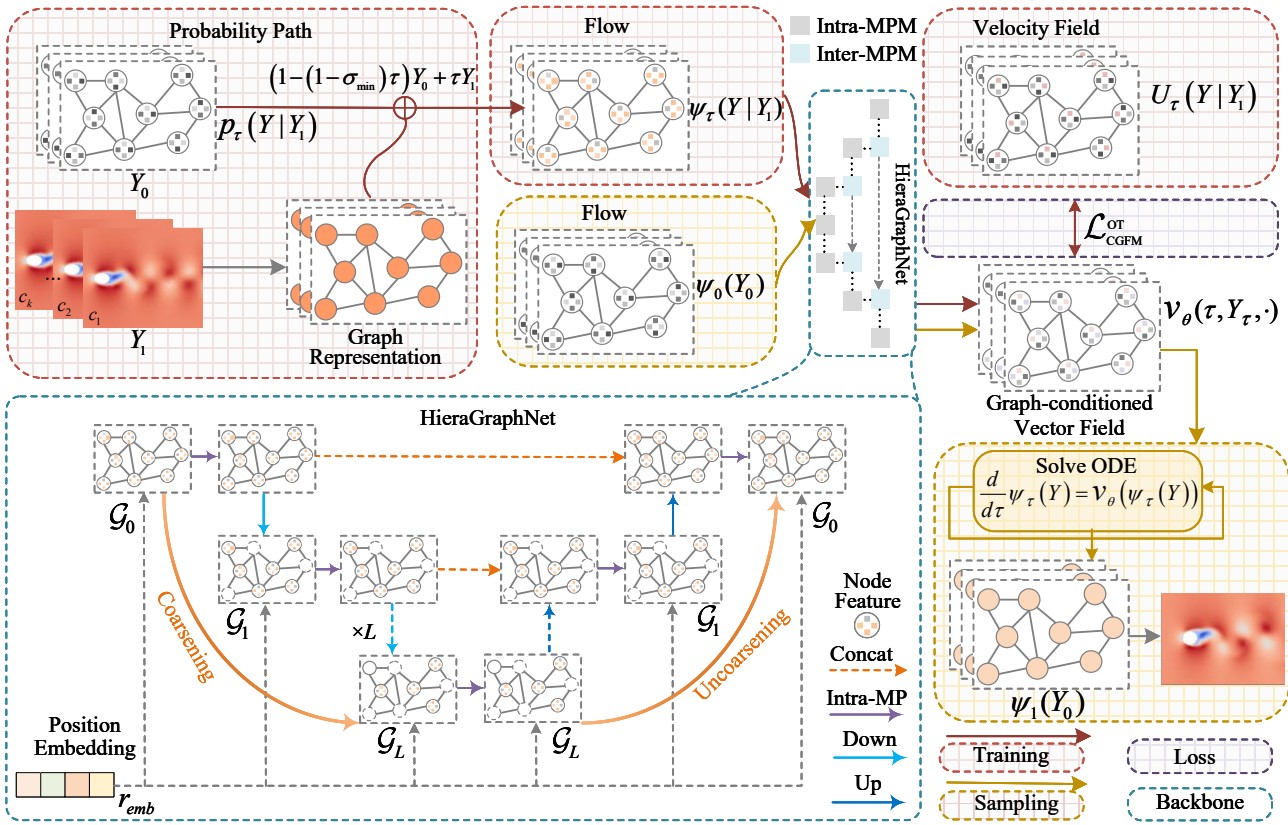

*Figure 2.* Overview of the entire architecture of the proposed CGFM. CGFM constructs a probability flow on the graph-node features of the discretized fluid domain and parameterizes the vector field conditioned on system-specific physical parameters using a hierarchical graph neural network (HieraGraphNet).

probability path $\{p_\tau(\cdot \mid \mathcal{G}, V_c, E_c)\}_{\tau \in [0,1]}$ from a tractable Gaussian prior $p_0$ to the equilibrium-state distribution $p_1(\cdot \mid \mathcal{G}, V_c, E_c)$. Afterward, we regress a graph-conditioned vector field $v_\theta(\tau, \cdot) : [0,1] \times \mathbb{R}^{|\mathcal{V}_\mathcal{M}| \times d} \rightarrow \mathbb{R}^{|\mathcal{V}_\mathcal{M}| \times d}$ to the target vector field $U_\tau$ such that its flow $\psi_\tau$ generates the desired probability path $p_\tau$. In general, the target vector field $U_\tau$ is intractable in closed form since it governs the joint transport between two high-dimensional sources and data distributions in complex fluid systems. To circumvent this intractability, we adopt the conditional optimal-transport path with a minimum noise scale $\sigma_{min}$ and use it to construct the intermediate state $Y_\tau$ via linear interpolation between the Gaussian prior sample $Y_0 \sim \mathcal{N}(0, I)$ and the target state sample $Y_1 \sim p_1(\cdot \mid \mathcal{G}, V_c, E_c)$. Using this linear path, for any $\tau \in [0,1]$, the intermediate state $Y_\tau$ and its corresponding target vector field $U_\tau$ are given by

$$Y_\tau = (1 - (1 - \sigma_{min})\tau)Y_0 + \tau Y_1 \sim p_\tau, \quad (2)$$

$$U_\tau := \frac{dY_\tau}{d\tau} = Y_1 - (1 - \sigma_{min})Y_0, \quad (3)$$

Under this construction, the target vector field $U_\tau$ is fully determined by the sampled endpoints $(Y_0, Y_1)$. Moreover, $U_\tau$ is independent of $\tau$ over $[0,1]$. Therefore, we train the

graph-conditioned vector field $v_\theta(\tau, \cdot)$ to approximate the target vector field $U_\tau$ on the mesh graph by minimizing the regression loss

$$\mathcal{L}_{\text{CGFM}}^{\text{OT}} = \mathbb{E}_{\tau, Y_1, Y_\tau} \left[ \| v_\theta(\tau, Y_\tau, \mathcal{G}, V_c, E_c) - U_\tau \|_2^2 \right], \quad (4)$$

where $\tau \sim \text{Unif}(0,1)$, $Y_1 \sim p_1$, $Y_\tau \sim p_\tau$. By jointly learning a graph-conditioned vector field across physical conditions, the model captures shared cross-system patterns and generalizes smoothly to unseen conditions. After training, for a given conditional graph information $(\mathcal{G}, V_c, E_c)$, we first draw an initial state sample $Y_0 \sim p_0 = \mathcal{N}(0, I)$, and then iteratively update it toward $\tau = 1$ according to the learned graph-conditioned velocity field $v_\theta$, i.e.,

$$Y_{\tau_{k+1}} = Y_{\tau_k} + \Delta\tau \cdot v_\theta(\tau_k, Y_{\tau_k}, \mathcal{G}, V_c, E_c), \quad (5)$$

where $k = 0, \ldots, K - 1$. The final state $Y_{\tau_K}$ is taken as the generated equilibrium-state sample, i.e., $Y_1 := Y_{\tau_K}$, which approximates a draw from the target distribution $p_1(\cdot \mid \mathcal{G}, V_c, E_c)$. Algorithm 2 describes the training and generation processes of the proposed method.

## 3.3. Learning the Graph-Conditioned Vector Field via HieraGraphNet

To generate plausible and diverse solution states, the prediction network is required to flexibly represent irregular geometries and adaptively allocate computations to regions with sharp spatial variations. The graph-based networks are well-suited for this purpose, as they can directly operate on unstructured simulation meshes. Motivated by this feature, CGFM parameterizes the graph-conditioned vector field $v_\theta$ with a HieraGraphNet, which takes the intermediate state $Y_\tau$ along the probability flow as input, and is conditioned on the graph $\mathcal{G}$, its node and edge features, and the flow step $\tau$

$$v_\theta \leftarrow \text{HieraGraphNet}(\tau, Y_\tau, \mathcal{G}, V_c, E_c). \quad (6)$$

Figure 2 illustrates the overall architecture of CGFM, a U-shaped hierarchical graph neural network built upon an encoder-propagator-decoder pipeline. First, the node encoder maps the concatenation of the noisy input features $\mathbf{v}_i^\tau$ with the conditional node features $\mathbf{v}_i^c$, $\mathbf{v}_i \leftarrow \text{mlp}_v([\mathbf{v}_i^\tau \,\|\, \mathbf{v}_i^c]) \in \mathbb{R}^{d_h}$, while the edge encoder maps the conditional edge features, $\mathbf{e}_{ij} \leftarrow \text{mlp}_e(\mathbf{e}_{ij}^c) \in \mathbb{R}^{d_h}$. Both encoders utilize a multilayer perceptron (mlp) to project their inputs into an $d_h$-dimensional latent space. The flow-step $\tau$ is encoded using a sinusoidal timestep embedding (Vaswani et al., 2017), followed by a linear layer and SELU activation function, which generates an $d_{emb}$-dimensional temporal embedding vector $\mathbf{r}_{\text{emb}}$. The temporal embedding $\mathbf{r}_{\text{emb}}$ is used to generate a pair of learnable parameters through an mlp, $(\boldsymbol{\gamma}, \boldsymbol{\beta}) \leftarrow \text{mlp}_{\gamma\beta}(\mathbf{r}_{\text{emb}})$, $\boldsymbol{\gamma}, \boldsymbol{\beta} \in \mathbb{R}^{d_h}$. These parameters are applied to linearly transform the encoded node features as $\tilde{\mathbf{v}}_i \leftarrow \boldsymbol{\gamma} \odot \mathbf{v}_i + \boldsymbol{\beta}$, where $\odot$ denotes element-wise multiplication. This allows the node features to adapt to the temporal embedding, thereby incorporating time-dependent information into the latent representations. The encoded node features $\{\tilde{\mathbf{v}}_i\}$, edge features $\{\mathbf{e}_{ij}\}$, together with the temporal embedding $\mathbf{r}_{\text{emb}}$ are then processed on $\mathcal{G}$ through a propagator,

$$(\{\mathbf{v}_i'\}, \{\mathbf{e}_{ij}'\}) \leftarrow \text{Prop}(\mathcal{G}, \{\tilde{\mathbf{v}}_i\}, \{\mathbf{e}_{ij}\}, \mathbf{r}_{\text{emb}}). \quad (7)$$

For large-scale fluid systems, the underlying dynamics involve long-range dependencies and multi-scale interactions that cannot be effectively captured by a single-resolution representation. Therefore, we design a hierarchical U-shaped structure that exchanges information both within each mesh resolution and across resolutions. Specifically, we define the original mesh graph $\mathcal{G}_0 := \mathcal{G}$ as the finest graph at level $L = 0$ and coarsen it $L$ times to automatically generate a sequence of lower-resolution graphs $\mathcal{G}_{1:L} = (\mathcal{V}_{1:L}, \mathcal{E}_{1:L})$, where $|\mathcal{V}_1| > |\mathcal{V}_2| > \cdots > |\mathcal{V}_L|$. At each level $\ell$, the propagator operates on a coarser graph $\mathcal{G}_\ell = (\mathcal{V}_\ell, \mathcal{E}_\ell)$ constructed through the TGC scheme. As shown in Figure 2, the propagator consists of two modules: (i) an *Intra-Level*

**Algorithm 1** Coarse node selection criterion

---

**Input:** graph $\mathcal{G}_\ell = (\mathcal{V}_\ell, \mathcal{E}_\ell)$, node coordinates $\{\mathbf{x}_i\}_{i \in \mathcal{V}_\ell}$
**Output:** $\text{mask}_\ell \in \{0, 1\}^{|\mathcal{V}_\ell|}$, coarse nodes $\mathcal{V}_{\ell+1}$
Initializing $\text{mask}_\ell[i] \leftarrow 1$ for all $i \in \mathcal{V}_\ell$
**for** each node $i \in \mathcal{V}_\ell$ **do**
  $d_i \leftarrow |\mathcal{N}_i^-|$
  $\rho_i \leftarrow \text{mean}_{j \in \mathcal{N}_i^-} \|\mathbf{x}_i - \mathbf{x}_j\|_2$
  $s_i \leftarrow d_i \cdot \rho_i$
**end for**
$\mathcal{I}_\ell \leftarrow \text{argsort}(\{s_i\}_{i \in \mathcal{V}_\ell})$   (in increasing order)
**for** each node $i \in \mathcal{I}_\ell$ **do**
  **if** $\text{mask}_\ell[i] = 1$ **then**
    **for** each node $j \in \mathcal{N}_i^-$ **do**
      $\text{mask}_\ell[j] \leftarrow 0$
    **end for**
  **end if**
**end for**
$\mathcal{V}_{\ell+1} \leftarrow \{ i \in \mathcal{V}_\ell \mid \text{mask}_\ell[i] = 1 \}$

---

*Message Passing Module (Intra-MPM)* and (ii) an *Inter-Level Message Passing Module (Inter-MPM)*. The implementation of the Intra- and Inter-MP Modules is provided in Appendix C. Given node and edge embeddings, information is first updated within $\mathcal{G}_\ell = (\mathcal{V}_\ell, \mathcal{E}_\ell)$ by two Intra-MP Modules, and then transferred to the lower-resolution graph $\mathcal{G}_{\ell+1}$ through an Inter-MP Module. This downsampling process is repeated $L - 1$ times until the features reach the lowest-resolution graph $\mathcal{G}_L$, followed by two Intra-MP Modules for global feature refinement. Subsequently, the features are upsampled through reverse Inter-MP modules and integrated with the corresponding intralevel features from the downsampling branch via skip connections. These fused representations are further refined by Intra-MP Modules at each level until the features are reconstructed on the original graph $\mathcal{G}_0$. Finally, the node embeddings at the finest resolution are decoded through a node-wise mlp to generate the target quantities.

**Topology- and Geometry-Aware Coarsening (TGC).** To generate graph hierarchies, we develop a topology- and geometry-aware coarsening (TGC) scheme that preserves both the geometric distribution and the topological connectivity of the mesh in coarser graphs. In this scheme, a subset of representative nodes is selected from the current graph $\mathcal{G}_\ell$ to create a coarser graph $\mathcal{G}_{\ell+1}$. Specifically, we assume that all the nodes of the input mesh are candidate coarse nodes and iteratively remove redundant nodes according to the node selection criterion. For each node $i \in \mathcal{V}_\ell$, we first compute its indegree $d_i = |\mathcal{N}_i^-|$, which represents the number of incoming neighbors, and its local geometric density

$$\rho_i = \frac{1}{|\mathcal{N}_i^-|} \sum_{j \in \mathcal{N}_i^-} \|\mathbf{x}_i - \mathbf{x}_j\|_2, \quad (8)$$

which measures the average spatial distance between the node and its incoming neighbors. A combined coarsening score is then defined as $s_i = d_i \cdot \rho_i$. Nodes with lower $s_i$ are prioritized as coarse nodes, as they tend to be spatially central within local neighborhoods while maintaining relatively low indegree. We traverse the nodes in ascending order of $s_i$. For each visited node, if it has not been dropped, we keep it as a coarse node and mark all of its in-neighbors as noncoarsen nodes. This procedure produces the coarsening mask $\text{mask}_\ell \in \{0, 1\}^{|\mathcal{V}_\ell|}$, from which the set of coarse nodes is obtained as $\mathcal{V}_{\ell+1} = \{ i \in \mathcal{V}_\ell \mid \text{mask}_\ell[i] = 1 \}$. The coarse node selection criterion is described in Algorithm 1. Once $\mathcal{V}_{\ell+1} \subset \mathcal{V}_\ell$ is obtained from $\mathcal{G}_\ell$, each fine node $i \in \mathcal{V}_\ell$ is assigned to a parent node $j \in \mathcal{V}_{\ell+1}$, denoted as

$$\mathcal{P}_i = \arg \min_{j \in \mathcal{V}_{\ell+1}} \left( \text{hop}(i,j), \|\mathbf{x}_i - \mathbf{x}_j\|_2 \right). \qquad (9)$$

Here, $\text{hop}(i,j)$ is the shortest-path distance on $\mathcal{G}_\ell$. If multiple parent candidates have the same hop distance, the one with the smallest Euclidean distance is chosen. For each coarse node $j \in \mathcal{V}_{\ell+1}$, the corresponding set of child nodes induced by $\mathcal{P}$ is defined as $\text{Ch}_j := \{ i \in \mathcal{V}_\ell \mid \mathcal{P}_i = j \}$. The edge set $\mathcal{E}_{\ell+1}$ of the coarsened graph $\mathcal{G}_{\ell+1} = (\mathcal{V}_{\ell+1}, \mathcal{E}_{\ell+1})$ is then constructed by preserving the connectivity patterns among the corresponding child nodes. Specifically, an edge $(j, q) \in \mathcal{E}_{\ell+1}$ is created if there exists at least one pair of child nodes $(i_1, i_2) \in \mathcal{E}_\ell$ with $i_1 \in \text{Ch}_j$ and $i_2 \in \text{Ch}_q$. All coarsening steps are performed as preprocessing steps and do not increase the runtime cost during training or inference.

### 3.4. Theoretical Analysis: Modeling Equilibrium Distributions from Short Trajectories

Learning equilibrium state distributions from short simulation trajectories is an ill-posed problem. In this subsection, we provide a theoretical justification showing that, under mild assumptions, jointly training across multiple systems enables the recovery and interpolation of conditional equilibrium distributions. Let $c := (\mathcal{G}, V_c, E_c) \in \mathcal{C}$ be the conditioning variable (e.g., mesh geometry, boundary conditions, and physical parameters), represented as conditional graph information. For each $c$, let $\pi_c$ denote the statistical-equilibrium distribution over converged snapshots $\mathbf{z} \in \mathcal{S}$. The training data consist of $N$ systems, where system $n$ provides a short equilibrium-stage sequence $\{\mathbf{z}_{n,t}\}_{t=t_0}^{t_0+m}$. Owing to temporal correlations, the effective number of independent samples can be approximated by

$$m_{n,\text{eff}} \approx \frac{m}{1 + 2 \sum_{k=1}^{m-1} \rho_n(k)}, \qquad (10)$$

where $\rho_n(k)$ denotes the lag-$k$ autocorrelation of the equilibrium process for system $n$. The total effective sample size across all the systems is $M_{\text{eff}} := \sum_{n=1}^{N} m_{n,\text{eff}}$. We define the population risk $\mathcal{R}(\theta) := \mathbb{E}_{c \sim P_C}[D(\pi_c, p_\theta(\cdot \mid c))]$,

where $D(\cdot, \cdot)$ denotes a discrepancy between distributions and $P_C$ represents the distribution across conditions. In practice, given short-trajectory data $\{(c_n, \{\mathbf{z}_{n,t}\}_{t=t_0}^{t_0+m})\}_{n=1}^{N}$, we minimize an empirical objective

$$\widehat{\mathcal{R}}(\theta) := \frac{1}{N} \sum_{n=1}^{N} \frac{1}{m} \sum_{t=t_0}^{t_0+m} \ell_\theta(\mathbf{z}_{n,t}, c_n), \qquad (11)$$

where $\ell_\theta$ represents a per-sample surrogate that is consistent with $D(\pi_c, p_\theta(\cdot \mid c))$. Our objective is to learn a conditional generative model $p_\theta(\cdot \mid c)$ that approximates $\pi_c$ across the condition space, thereby enabling sampling from $\pi_{c^*}$ under an unseen condition $c^*$. To formalize this, we introduce two assumptions that correspond to the key capabilities of our model: *shared structure across systems* and *interpolation across the condition space*.

**Assumption 3.1** (Shared structure). There exists a model class $\{p_\theta(\cdot \mid c) : \theta \in \Theta\}$ that can approximate the family of conditional equilibrium distributions $\{\pi_c : c \in \mathcal{C}\}$ in $P_C$-average. Formally, there exists $\varepsilon \geq 0$ such that $\inf_{\theta \in \Theta} \mathbb{E}_{c \sim P_C}[D(\pi_c, p_\theta(\cdot \mid c))] \leq \varepsilon$.

**Assumption 3.2** (Interpolability). There exists a metric $d_\mathcal{C}$ on $\mathcal{C}$ and a constant $L > 0$ such that, for all $c, c' \in \mathcal{C}$, the mapping $c \mapsto \pi_c$ is Lipschitz continuous under the distributional discrepancy $D$, i.e., $D(\pi_c, \pi_{c'}) \leq L\, d_\mathcal{C}(c, c')$.

Based on Assumptions 3.1–3.2, we now focus on two analytical objectives. In the training stage, each system provides only a short equilibrium trajectory. Thus, it is necessary to analyze whether, under the shared-structure assumption, joint training can reduce the estimation error according to the total effective sample size $M_{\text{eff}}$, thereby mitigating the ill-posedness of individual systems. In addition, we need to perform generation and statistical estimation for unseen conditions $c^*$. Under the interpolability assumption, we characterize whether the approximation learned by the model under training conditions can be generalized to unseen conditions, and how this generalization error is controlled by the coverage of training conditions in the condition space. Therefore, we present Theorem 3.3 and Theorem 3.4.

**Theorem 3.3** (Cross-system aggregation reduces estimation error). *Let $\widehat{\theta}$ be an empirical risk minimizer of $\widehat{\mathcal{R}}(\theta)$. Then, for any $\delta \in (0, 1)$, with a probability of at least $1 - \delta$,*

$$\mathcal{R}(\widehat{\theta}) \leq \inf_{\theta \in \Theta} \mathcal{R}(\theta) + O\left(\frac{\mathfrak{C}(\Theta)}{\sqrt{M_{\text{eff}}}}\right) + O\left(\sqrt{\frac{\log(1/\delta)}{M_{\text{eff}}}}\right),$$

*where $\mathfrak{C}(\Theta)$ denotes a model-complexity term.*

Theorem 3.3 characterizes the origin of the apparent ill-posedness and reveals its alleviation mechanism. For a *single* system, the estimation error typically scales as $O(1/\sqrt{m_{n,\text{eff}}})$, which leads to unstable estimation when

the trajectory length is short. In contrast, with shared parameters (Assumption 3.1), joint training aggregates information across systems and yields an error scaling as $O(1/\sqrt{M_{\text{eff}}})$. This result indicates that the estimation error is controlled by the total effective sample size $M_{\text{eff}}$ and thus decreases as the number of systems $N$ increases. Then, we further define the coverage radius of the condition space as $\rho_N = \sup_{c \in \mathcal{C}} \min_{n \in [N]} d_{\mathcal{C}}(c, c_n)$. As the number of training systems $N$ increases, the coverage of the condition space becomes denser and the coverage radius $\rho_N$ decreases.

**Theorem 3.4** (Interpolation bound for unseen conditions). *Under Assumptions 3.1–3.2, the learned model $q_{\hat{\theta}}$ achieves a uniform training discrepancy of at most $\epsilon$ under the training conditions, i.e., $\max_{n \in [N]} D\big(\pi_{c_n}, p_{\hat{\theta}}(\cdot \mid c_n)\big) \leq \epsilon$. Then, for any unseen condition $c^*$, it holds that $D\big(\pi_{c^*}, p_{\hat{\theta}}(\cdot \mid c^*)\big) \leq \epsilon + \varepsilon + L\rho_N$.*

Theorem 3.4 indicates that the discrepancy under an unseen condition is controlled by the sum of a training-driven estimation term $\epsilon$, a model approximation term $\varepsilon$ from Assumption 3.1, and a coverage-induced interpolation term $L\rho_N$ that decreases as the training conditions cover $\mathcal{C}$ more densely. In conjunction with Theorem 3.3, this provides a theoretical justification for learning to sample equilibrium states from short trajectories by leveraging multiple systems and interpolating across the condition space.

# 4. Experiments

## 4.1. Experimental Settings

We conduct our experiments using three canonical dynamical scenarios introduced by (Lino et al., 2022) and (Valencia et al., 2025), which are publicly available and widely used as benchmarks for learning-based fluid simulations. Detailed descriptions of the datasets are provided in Appendix B. We compare our model with representative baseline methods from three categories. 1) Deterministic models: MuGS-GNN (Lino et al., 2022), Geo-FNO (Li et al., 2023b), Transolver (Wu et al., 2024). 2) Probabilistic models: Bayesian GNN (Chen et al., 2023), Gaussian regression GNN (GR-GNN) (Li et al., 2023a), and Gaussian mixture graph U-Net (GM-GUN) (Yang et al., 2024). 3) Generative models: Variational graph autoencoder (VGAE) (Su et al., 2025), Diffusion graph network (DGN) (Valencia et al., 2025), and Graph flow matching (GFM) (Siddiqui et al., 2026). We use the mean coefficient of determination ($R^2$), node-wise and graph-wise Wasserstein-2 distances ($W_2^{\text{node}}$ and $W_2^{\text{graph}}$) to validate the model performance, which is consistent with (Lino et al., 2022; Valencia et al., 2025). The metrics are described in detail in Appendix E. For these metrics, a higher $R^2$ value indicates a closer match to the ground truth, while lower $W_2^{\text{node}}$ and $W_2^{\text{graph}}$ values reflect better distributional accuracy.

## 4.2. Performance Evaluation

**Learning the distributions from incomplete trajectories.** We first evaluate whether the model can learn the probability distribution of converged flow states from incomplete training trajectories in canonical 2D flow problems. Figure 6 shows the probability density function (PDF) for a system from the Ellipse-Indist dataset. We can see that the deterministic model fails to recover the PDF of the ground truth, while the generative models approximate it more faithfully, with CGFM yielding a PDF that most closely matches the ground truth. Tables 1 and 10 summarize the node-wise and graph-wise Wasserstein-2 distances for the Ellipse and EllipseFlow tasks, respectively. Although MuGS-GNN attains competitive $R^2$ values, its Wasserstein-2 distances remain significantly greater. This result reflects that deterministic models fit individual snapshots well but fail to recover the underlying probability of flow states. In contrast, CGFM consistently achieves the smallest Wasserstein-2 distances among all the baselines and maintains the highest $R^2$ across all the datasets. These results indicate that CGFM is capable of effectively extracting the shared patterns across multiple conditions while accurately learning the full equilibrium-state distribution from incomplete trajectories. Additional experimental details are presented in Appendix F.1.

**Superior generative performance of CGFM.** We evaluate the generative performance of different models on the Ellipse(Flow) dataset from two perspectives: (i) qualitative visual comparisons of the generated flow fields, and (ii) quantitative accuracy measured by the $R^2$. As illustrated in Figure 3, compared with the baseline approaches, CGFM can generate pressure field predictions on the surface of an ellipse that is visually closest to the ground truth for unobserved physical conditions. Figure 7 shows velocity field samples around an elliptical cylinder, highlighting that compared with the competing approaches, CGFM generates more coherent and physically realistic flow structures. Figure 8 further shows the same advantage for out-of-distribution datasets in the EllipseFlow task. Although the DGN achieves competitive $R^2$ values, it suffers from undesirable high-frequency noise in the generated samples. These observations are also confirmed by the quantitative results in Tables 8 and 9, where our model achieves the highest $R^2$ values across all the test datasets.

**Generalization to large-scale irregular 3D meshes.** To assess the ability of CGFM to generalize to large, irregular geometries, we evaluate it on the Wing task, a 3D turbulent-flow benchmark whose surface meshes contain approximately 7,000 nodes per sample. As shown in Figures 5 and 10, the pressure fields generated by our model are visually closest to the ground truth, particularly in regions with strong three-dimensional effects such as the leading

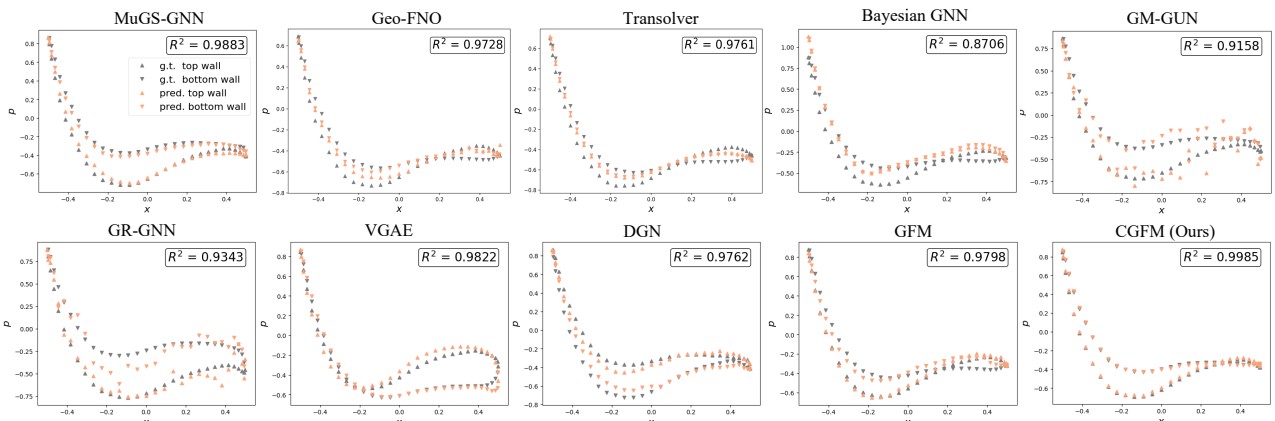

*Figure 3.* Pressure field predictions on the surface of an ellipse from different models for a system in the Ellipse-Indist dataset.

*Table 1.* Mean node-wise ($W_2^{\text{node}}$) and graph-wise ($W_2^{\text{graph}}$) Wasserstein-2 distances of all the models on the Ellipse datasets.

| | $W_2^{\text{node}} \downarrow$ | | | | | $W_2^{\text{graph}} \downarrow$ | | | | |
|---|---|---|---|---|---|---|---|---|---|---|
| | -InDist | -Thin | -Thick | -LowRe | -HighRe | -InDist | -Thin | -Thick | -LowRe | -HighRe |
| MuGS-GNN | 0.1078 | 0.0424 | 0.2290 | 0.0801 | 0.1457 | 0.9727 | 0.3621 | 2.1386 | 0.7235 | 1.2979 |
| Geo-FNO | 0.0799 | 0.0467 | 0.1931 | 0.0577 | 0.1111 | 0.7930 | 0.4465 | 1.8959 | 0.5916 | 1.0673 |
| Transolver | 0.0748 | 0.0329 | 0.1748 | 0.0544 | 0.1048 | 0.8003 | 0.4017 | 1.7946 | 0.6109 | 1.0634 |
| Bayesian GNN | 0.1254 | 0.1692 | 0.1155 | 0.1459 | 0.1044 | 1.7784 | 1.6091 | 2.4126 | 1.7462 | 1.8084 |
| GM-GNN | 0.0618 | 0.0179 | 0.1296 | 0.0428 | 0.0854 | 1.0945 | 0.4466 | 2.2971 | 0.7905 | 1.4780 |
| GR-GNN | 0.0670 | 0.0226 | 0.1534 | 0.0436 | 0.0907 | 1.0734 | 0.4471 | 2.2926 | 0.7469 | 1.4462 |
| VGAE | 0.0350 | 0.0227 | 0.0718 | 0.0271 | 0.0462 | 0.6141 | 0.2804 | 1.3974 | 0.4871 | 0.7993 |
| DGN | 0.0445 | 0.0285 | 0.0923 | 0.0296 | 0.0574 | 0.4523 | 0.2824 | 0.9235 | 0.3261 | 0.5917 |
| GFM | 0.0322 | 0.0190 | 0.0706 | 0.0229 | 0.0458 | 0.5852 | 0.2307 | 1.3307 | 0.4411 | 0.7646 |
| CGFM (Ours) | **0.0239** | **0.0164** | **0.0602** | **0.0166** | **0.0372** | **0.2734** | **0.1574** | **0.6592** | **0.1944** | **0.4415** |

*Table 2.* Quantitative results in terms of $R^2$, $W_2^{\text{node}}$, and $W_2^{\text{graph}}$ on the Wing dataset.

| | $R^2 \uparrow$ | $W_2^{\text{node}} \downarrow$ | $W_2^{\text{graph}} \downarrow$ |
|---|---|---|---|
| GM-GUN | 0.9490 ± 0.0282 | 0.0259 | 4.2996 |
| VGAE | 0.9851 ± 0.0058 | 0.0255 | 2.6993 |
| DGN | 0.9795 ± 0.0329 | 0.0178 | 2.3831 |
| GFM | 0.9652 ± 0.0214 | 0.0327 | 3.8579 |
| CGFM (Ours) | **0.9896 ± 0.0098** | **0.0099** | **1.9265** |

*Table 3.* Comparison of model efficiency on the Wing dataset.

| | GPU Memory (GiB) | Running time (s/epoch) | Inference time (s/sample) |
|---|---|---|---|
| DGN | 1.57 | 66.49 | 0.1377 |
| GFM | 3.80 | 88.55 | 0.1759 |
| **CGFM** | **1.59** | **65.36** | **0.0981** |

edge and wing tip, whereas the baseline models tend to either oversmooth or distort regions with strong pressure gradients. As shown in Table 2, CGFM achieves the best overall balance of sample fidelity, node-wise distribution accuracy and spatially coherent distribution accuracy among all the approaches. This finding demonstrates that CGFM not only accurately predicts individual turbulent realizations, but also captures the underlying distribution of system states on large, irregular 3D meshes.

**Efficiency Analysis.** To assess the efficiency of generative modeling on large-scale 3D meshes, we compare the DGN, GFM, and CGFM on the Wing dataset in terms of training time per epoch and inference time for generating 3,000

samples. All efficiency measurements are performed on a single NVIDIA GeForce RTX 4090 GPU (24 GB). Running time is measured by the time to complete one epoch, which contains 1000 iterations. For inference efficiency, we report the wall-clock time required to generate 3,000 samples for each generative model. To ensure a fair comparison, all models are implemented with comparable architectural configurations, with approximately 5.4M trainable parameters each. As shown in Table 3, CGFM achieves superior computational efficiency among the compared generative models. It reduces the inference time by approximately 28.8% and 44.2% compared with DGN and GFM, respectively, while maintaining GPU memory usage comparable to DGN and substantially lower than GFM.

We further compare the computational efficiency of generative sampling with that of a traditional numerical solver.

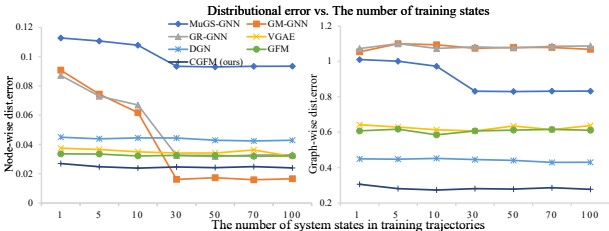

*Figure 4.* Ablation study of distributional error with varying numbers of states per training trajectory on the Ellipse-Indist dataset.

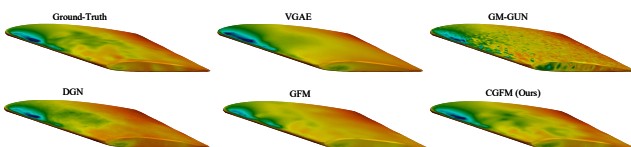

*Figure 5.* Pressure fields ($p$) generated by the VGAE, GM-GUN, DGN, GFM, and CGFM for a simulation on the Wing test dataset.

Traditional solvers must evolve the system through a long transient regime before obtaining equilibrium states for statistical estimation. In contrast, CGFM directly samples from the learned equilibrium distribution and thus avoids the costly transient simulation stage. A further practical advantage of CGFM is that, unlike the traditional solver (official OpenFOAM distributions lack GPU support), it can be executed efficiently on GPUs. As shown in Table 12, CGFM generates 3,000 samples in 185.372 minutes on a CPU and 4.91 minutes on a GPU, which correspond to speed-ups of 16.05× and 606.11×, respectively, over the numerical solver. Detailed solver settings, CPU/GPU inference times, and speed-up comparisons are provided in Appendix F.2.

### 4.3. Ablation Studies

To validate the effects of key components in our method, we conduct ablation studies, including (i) varying the number of states per training trajectory, (ii) removing the hierarchical message passing and operating only on the finest mesh $\mathcal{G}_0$ (w/o hierarchy), (iii) replacing the TGC scheme with Guillard's coarsening algorithm, (iv) varying the number of hierarchy levels in HieraGraphNet, and (v) varying the number of inference steps. Details are provided in Appendix F.3.

Figure 4 shows the distributional error of our model and the baselines on the Ellipse-Indist dataset. Our model exhibits the least sensitivity to the number of states per training trajectory. To verify the effectiveness of HieraGraphNet, we construct a single-scale variant of CGFM that operates only on the finest graph $\mathcal{G}_0$. As shown in Tables 4 and 13–14, the single-scale variant of CGFM shows markedly worse performance across all the metrics on both the EllipseFlow task (approximately 2.3k nodes per graph) and the Wing task (approximately 6.8k nodes per graph). We evaluate the contribution of the proposed TGC scheme by replacing it

*Table 4.* Comparison of CGFM and its variants in terms of $R^2$, $W_2^{\text{node}}$, and $W_2^{\text{graph}}$ on the Wing dataset.

| | $R^2 \uparrow$ | $W_2^{\text{node}} \downarrow$ | $W_2^{\text{graph}} \downarrow$ |
|---|---|---|---|
| CGFM (Ours) | **0.9896 ± 0.0098** | **0.0099** | **1.9265** |
| w/o hierarchy | 0.9712 ± 0.0199 | 0.0263 | 3.5026 |
| CGFM w/ Guillard | 0.9767 ± 0.0348 | 0.0141 | 2.3697 |

with Guillard's coarsening. As shown in Table 4, substituting TGC with the alternative coarsening strategy leads to consistent decreases in both the sample accuracy and distributional accuracy. In particular, the CGFM w/ Guillard variant exhibits higher $W_2^{\text{node}}$ and $W_2^{\text{graph}}$, which indicates that it fails to preserve the multi-scale spatial correlations required for modeling turbulent pressure distributions. The visual comparisons in Figure 11 further illustrate these advantages. Moreover, we evaluate the robustness of CGFM under genuine topology changes on the CruiseAOA=5° subset of TandemFoilSet, where CGFM w/ TGC consistently outperforms the CGFM w/ Guillard variant in a challenging multi-body aerodynamic setting. To assess the effect of hierarchy depth, we compare CGFM variants with different numbers of hierarchy levels in Table 16. As the number of hierarchy levels increases, the performance consistently improves in terms of $R^2$, $W_2^{\text{node}}$ and $W_2^{\text{graph}}$. These results indicate that deeper hierarchies better capture multi-scale representations and long-range interactions. In addition, we conduct a sensitivity study on the Wing dataset by varying the number of inference steps and evaluating $W_2^{\text{node}}$ computed from 3,000 generated samples. The results show that CGFM can maintain high distributional accuracy with very few inference steps.

## 5. Conclusions

We introduced a condition-aware graph flow matching method that learns the full equilibrium-state distribution of irregular fluid systems from incomplete training trajectories across multiple conditions. By coupling the efficient distributional modeling capability of condition-aware flow matching with the geometric flexibility of HieraGraphNet, CGFM can effectively capture shared patterns and dynamic dependencies across different trajectories, which in turn allows robust interpolation and extrapolation to unseen conditions in large-scale fluid systems with complex geometries. The topology- and geometry-aware coarsening scheme adaptively retains nodes in regions of high geometric density and structural complexity, thereby generating a more uniform relative node distribution within the domain. Our experimental results on 2D and 3D fluid dynamics demonstrated that compared with the baseline models, CGFM achieves state-of-the-art results, generating high-fidelity flow fields while reproducing the complete underlying distribution of converged states.

## Acknowledgments

This work was supported in part by National Natural Science Foundation of China (NSFC) under Grant 62576231, in part by the Natural Science Foundation of Sichuan Province under Grant 2026NSFSC0420, and in part by the Open Funding of National Key Laboratory of Fundamental Algorithms and Models for Engineering Simulation.

## Impact Statement

This paper aims to advance machine learning for modeling and sampling equilibrium states in fluid systems. While the proposed method may improve the efficiency of scientific simulation and analysis, we do not anticipate societal or ethical impacts beyond those commonly associated with general-purpose generative modeling and surrogate modeling, and thus none are specifically highlighted here.

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

## A. Flow Matching for Distribution Modeling

Flow matching (FM) has recently emerged as a compelling alternative to diffusion models, which offers a more stable and computationally efficient framework for generative modeling (Lipman et al., 2023; Kornilov et al., 2024). Given a source distribution $p_0$ (e.g., Gaussian noise) and a data distribution $p_1$, FM learns a time-dependent vector field $v_\theta(Y_\tau, \tau)$, parameterized by a neural network, that transports $p_0$ into $p_1$ through an interpolating probability path $\{p_\tau\}_{\tau \in [0,1]}$. The learning objective is defined as follows:

$$\mathcal{L}_{\text{FM}}(\theta) = \mathbb{E}_{\tau \sim \mathcal{U}[0,1], \, Y \sim p_\tau} \big\| v_\theta(Y_\tau, \tau) - u(Y_\tau, \tau) \big\|^2, \tag{12}$$

where $\theta$ denotes the model parameters and $u(Y_\tau, \tau)$ represents the target velocity field. An intuitive and effective formulation leverages optimal transport to define a conditional flow that linearly transports samples from the source distribution $p_0 \sim \mathcal{N}(0, I)$ to the data distribution $p_1$:

$$u_\tau(Y \mid Y_1) = \frac{Y_1 - (1 - \sigma_{\min})Y}{1 - (1 - \sigma_{\min})\tau}, \tag{13}$$

where $\sigma_{\min}$ is introduced to avoid numerical instability. The corresponding interpolation is given by

$$\psi_\tau(Y) = (1 - (1 - \sigma_{\min})\tau)Y + \tau Y_1. \tag{14}$$

This approach yields a straight-line conditional flow with a time-independent vector field. The flexibility of constructing probability paths further allows flow matching to incorporate physical priors and domain-specific constraints, making it especially suitable for modeling complex dynamical systems such as fluid flows.

## B. Datasets

Three fluid simulation datasets, EllipseFlow, Ellipse, and Wing are used in the experiments. The simulation details can be found in Tables 5–6. The datasets are described as follows:

- **EllipseFlow:** The training and testing datasets are low-resolution versions of the datasets from (Lino et al., 2022), which contain two-dimensional unsteady laminar flow simulations of velocity and pressure fields around an elliptical cylinder parallel to the free-stream flow. The flow exhibits periodic vortex shedding resulting from an unstable low-pressure wake behind the cylinder. The training dataset comprises 5,000 simulations, each consisting of 101 consecutive time steps of fully developed flow, with Reynolds numbers ($Re$) in the range of 500-1,000 and the relative thickness of ellipses ($b$) between 0.5 and 0.8. Each test dataset comprises 50 simulations and is used to evaluate the model's interpolation capabilities for in-distribution $Re$ and $b$, as well as its extrapolation capabilities for unseen values of $Re$ and $b$. The parameters of each dataset are listed in Table 5.

- **Ellipse:** A surface-only variant of the EllipseFlow dataset. The Ellipse dataset focuses solely on inferring the pressure distribution on the surface of the elliptical cylinder. It shares the same simulation parameters ($Re$ and relative thickness ratios) as the EllipseFlow dataset (see Table 5).

- **Wing:** A three-dimensional turbulent flow simulation dataset, aimed at predicting the surface pressure distribution on wings, was generated by (Valencia et al., 2025) using Detached Eddy Simulation (DES) via OpenFOAM's PISO solver, with mesh geometries constructed using snappyHexMesh. The simulations were conducted under a free-stream velocity of 100 km/h and a kinematic viscosity of $1.5 \times 10^{-5}$ m$^2$/s, which correspond to a high Reynolds number ($\sim 2 \times 10^6$). The wing geometries in this dataset vary in terms of relative thickness, taper ratio, sweep angle, and twist angle. The training dataset consists of 1,000 simulations, each with 250 time steps, while the test dataset comprises 16 longer simulations, each with 2,500 time steps. The details of the parameter values for the dataset are listed in Table 6.

The Ellipse, EllipseFlow, and Wing datasets use different input conditions for node and edge features. In the Ellipse task, each node encodes the Reynolds number ($Re$) as a condition, while each edge represents the relative position between adjacent nodes ($\mathbf{x}_j - \mathbf{x}_i$) and the projection of the free-stream velocity along the edge ($\mathbf{u}_\infty \cdot \hat{\mathbf{t}}_{ij}$). The output at each node is the pressure ($p_i$). For the EllipseFlow task, each node condition includes the Reynolds number ($Re$) and a one-hot vector ($\boldsymbol{\omega}_i$) indicating the node type (inlet, wall, or inner), the edge condition represents the relative position between nodes

*Table 5.* Parameter values of the Ellipse and EllipseFlow systems in the datasets.

| Dataset | $Re$ | $b$ | No. Simulations | Train/Test |
|---------|------|-----|-----------------|------------|
| Ellipse(Flow)-Train | 500-1000 | 0.5-0.8 | 5000 | Training |
| Ellipse(Flow)-Indist | 500-1000 | 0.5-0.8 | 50 | Testing |
| Ellipse(Flow)-LowRe | 400-500 | 0.5-0.8 | 50 | Testing |
| Ellipse(Flow)-HighRe | 1000-1100 | 0.5-0.8 | 50 | Testing |
| Ellipse(Flow)-Thin | 500-1000 | 0.45-0.5 | 50 | Testing |
| Ellipse(Flow)-Thick | 500-1000 | 0.8-0.9 | 50 | Testing |

*Table 6.* Parameter values of the Wing systems in the training and test datasets.

| Dataset | Thickness | Taper ratio | Sweep (degrees) | Twist (degrees) | No. Simulations | Train/Test |
|---------|-----------|-------------|-----------------|-----------------|-----------------|------------|
| Wing-Train | [10,14] | [0.3,0.7] | [20,40] | [-5,5] | 1000 | Training |
| Wing-Test | 11,13 | 0.4,0.6 | 25,35 | -2.5,2.5 | 16 | Testing |

$(\mathbf{x}_j - \mathbf{x}_i)$. The predicted outputs include the velocity components $(u_i, v_i)$ and the pressure $(p_i)$ at each node. The Wing dataset involves nodes with a unit outer normal vector $(\hat{\mathbf{n}}_i)$, and each edge encodes the relative position between nodes $(\mathbf{x}_j - \mathbf{x}_i)$, along with several projections of the free-stream velocity $(\mathbf{u}_\infty)$. These projections include the velocity along the edge direction $(\mathbf{u}_\infty \cdot \hat{\mathbf{t}}_{ij})$, the normal direction $(\mathbf{u}_\infty \cdot \hat{\mathbf{n}}_{ij})$, and the binormal direction $(\mathbf{u}_\infty \cdot \hat{\mathbf{b}}_{ij})$, where $\hat{\mathbf{b}}_{ij} := \hat{\mathbf{t}}_{ij} \times \hat{\mathbf{n}}_i$. The output is the pressure $(p_i)$ at each node. The input encodings for mesh edges $\mathbf{e}_{ij}$ and nodes $\mathbf{v}_i$, as well as the predicted outputs for each system, are provided in Table 7.

*Table 7.* The conditional inputs and their predicted outputs for each system.

| Dataset | Node conditions $\mathbf{v}_i$ | Edge conditions $\mathbf{e}_{ij}$ | Node outputs |
|---------|-------------------------------|-----------------------------------|--------------|
| Ellipse | $Re$ | $\mathbf{x}_j - \mathbf{x}_i, \mathbf{u}_\infty \cdot \hat{\mathbf{t}}_{ij}$ | $p_i$ |
| EllipseFlow | $Re, \boldsymbol{\omega}_i$ | $\mathbf{x}_j - \mathbf{x}_i$ | $u_i, v_i, p_i$ |
| Wing | $\hat{\mathbf{n}}_i$ | $\mathbf{x}_j - \mathbf{x}_i, \mathbf{u}_\infty \cdot \hat{\mathbf{t}}_{ij}, \mathbf{u}_\infty \cdot \hat{\mathbf{n}}_{ij}, \mathbf{u}_\infty \cdot \hat{\mathbf{b}}_{ij}$ | $p_i$ |

## C. Intra- and Inter-MP Modules

We provide the detailed update rules of the Intra-Level and Inter-Level Message Passing Modules described in Section 3.3.

**Intra-MP Module.** The Intra-Level Message Passing Module in the propagator follows the general framework described in (Battaglia et al., 2016; 2021). At each resolution level, the node and edge features are iteratively updated through edge-update, edge-aggregation, and node-update steps as follows:

$$\mathbf{e}_{ij} \leftarrow \mathbf{W}_e \mathbf{e}_{ij} + \phi_\theta^e\big(\text{LN}([\mathbf{e}_{ij} \| \mathbf{v}_i \| \mathbf{v}_j])\big), \tag{15}$$

$$\bar{\mathbf{e}}_j \leftarrow \sum_{i \in \mathcal{N}_j^-} \mathbf{e}_{ij}, \tag{16}$$

$$\mathbf{v}_j \leftarrow \mathbf{W}_v \mathbf{v}_j + \phi_\theta^v\big(\text{LN}([\bar{\mathbf{e}}_j \| \mathbf{v}_j])\big), \tag{17}$$

where $\text{LN}(\cdot)$ denotes layer normalization, and $\phi_\theta$ represents the MLPs with one hidden layer of $d_h$ neurons. Repeated application of Eqs. (15)–(17) expands the receptive field of each node and allows local physical interactions to be integrated effectively within each scale.

**Inter-MP Module.** The Inter-Level Message Passing Module propagates information across adjacent graph hierarchies through downsampling and upsampling operations. During the downsampling phase (fine-to-coarse), each parent node $j \in \mathcal{V}_{\ell+1}$ aggregates messages from its children $i \in \text{Ch}_j \subset \mathcal{V}_\ell$ according to

$$\mathbf{v}_j = \sum_{i \in \text{Ch}_j} \phi_\theta\big(\text{LN}([\text{Linear}(\mathbf{x}_j - \mathbf{x}_i) \| \mathbf{v}_i])\big), \quad \forall j \in \mathcal{V}_{\ell+1}. \tag{18}$$

---

**Algorithm 2** Condition-Aware Graph Flow Matching

---

**Training Process**

**Require** The mesh-node states $Y_1$, system graph $\mathcal{G}$ and its node/edge attributes $(V_c, E_c)$, and initial network $v_\theta$, constant $\sigma_{min}$.

**while** Training **do**

    Sampling $Y_0 \sim p_0, Y_1 \sim p_1, \tau \sim \text{Uniform}[0, 1]$

    Interpolating $Y_\tau \leftarrow (1 - (1 - \sigma_{min})\tau)Y_0 + \tau Y_1$

    Computing true velocity $U_\tau \leftarrow Y_1 - (1 - \sigma_{min})Y_0$

    Predicting $v_\theta \leftarrow \text{HieraGraphNet}(\tau, Y_\tau, \mathcal{G}, V_c, E_c)$

    Computing $\mathcal{L}_{\text{CGFM}}^{\text{OT}} \leftarrow \|v_\theta(\tau, Y_\tau, \cdot) - U_\tau\|^2$

    $\theta \leftarrow \text{Update}(\theta, \nabla_\theta \mathcal{L}_{\text{CGFM}}^{\text{OT}}(\theta))$

**end while**

**Euler Sampling**

**Require** System graph $\mathcal{G}$ and its attributes $(V_c, E_c)$, trained vector field $v_\theta(\tau, \cdot)$, number of ODE steps $K$.

Sampling $Y_0 \sim \mathcal{N}(0, I)$

Setting $\Delta\tau \leftarrow 1/K, \tau_k \leftarrow k\Delta\tau$ for $k = 0, \ldots, K$

Initializing $Y_{\tau_0} \leftarrow Y_0$

**for** $k = 0$ **to** $K - 1$ **do**

    $Y_{\tau_{k+1}} \leftarrow Y_{\tau_k} + \Delta\tau \, v_\theta(\tau_k, Y_{\tau_k}, \mathcal{G}, V_c, E_c)$

**end for**

Output the generated state $Y_1 \leftarrow Y_{\tau_K}$

---

The edge features $\mathcal{E}_{\ell+1}$ are defined as $\mathbf{e}_{jq} = \text{Linear}(\mathbf{x}_j - \mathbf{x}_q), \quad \forall(j, q) \in \mathcal{E}_{\ell+1}$. In the upsampling phase (coarse-to-fine), each node $i \in \mathcal{V}_{\ell-1}$ receives information from its parent $j = \mathcal{P}_i \in \mathcal{V}_\ell$ as

$$\mathbf{v}_i = \phi_\theta\big(\text{LN}([\text{Linear}(\mathbf{x}_i - \mathbf{x}_j)\|\mathbf{v}_j\|\mathbf{v}_i])\big), \quad \forall i \in \mathcal{V}_{\ell-1}. \tag{19}$$

The edge features $\mathcal{E}_{\ell-1}$ are obtained from the downsampling branch via a skip connection. These inter-level message-passing operations allow information to flow bidirectionally between coarse and fine resolutions. The U-Net structure enables the propagator to capture both multi-scale dynamics and long-range dependencies, while effectively filtering high- and low-frequency noise (Si et al., 2024).

## D. CGFM Algorithm

Algorithm 2 presents the overall pipeline of Condition-aware Graph Flow Matching (CGFM), including both training and sampling. During training, CGFM learns a graph-conditioned vector field by drawing a prior sample $Y_0 \sim p_0$, a target equilibrium-state sample $Y_1 \sim p_1$, and a flow time $\tau \sim \text{Uniform}[0, 1]$. It then constructs an intermediate state $Y_\tau$ along a prescribed optimal-transport path and computes the corresponding ground-truth velocity $U_\tau$. Conditioned on the system graph $\mathcal{G}$ and its node/edge attributes $(V_c, E_c)$, HieraGraphNet predicts the graph-conditioned vector field $v_\theta(\tau, Y_\tau, \mathcal{G}, V_c, E_c)$, and the model parameters are optimized by minimizing an $\mathcal{L}_{\text{CGFM}}^{\text{OT}}$ regression loss. At inference time, CGFM generates equilibrium-state samples by integrating the learned probability-flow ODE. Starting from Gaussian noise $Y_0 \sim \mathcal{N}(0, I)$, our method performs Euler updates over $K$ uniform time steps, iteratively advancing the state using the learned vector field conditioned on $(\mathcal{G}, V_c, E_c)$. The final iterate $Y_{\tau_K}$ is returned as the generated sample $Y_1$.

## E. Evaluation Metrics

We use the mean coefficient of determination ($R^2$), node-wise and graph-wise Wasserstein-2 distances ($W_2^{\text{node}}$ and $W_2^{\text{graph}}$) to validate the performance of the model. Specifically, we measure the sample accuracy by calculating the mean coefficient of determination ($R^2$) between the generated fields and their ground-truth equivalents, which is consistent with (Lino et al., 2022; Valencia et al., 2025). For a system state $\mathbf{s}_{i,t}$ defined on the mesh nodes $\mathcal{V}_\mathcal{M}$ at time step $t$, the coefficient of determination is computed as follows:

$$R^2 = 1 - \frac{\sum_{i \in N} \left(\mathbf{s}_{i,t}^{\text{gt}} - \mathbf{s}_{i,t}\right)^2}{\sum_{i \in N} \left(\mathbf{s}_{i,t}^{\text{gt}} - \bar{\mathbf{s}}_t^{\text{gt}}\right)^2}. \tag{20}$$

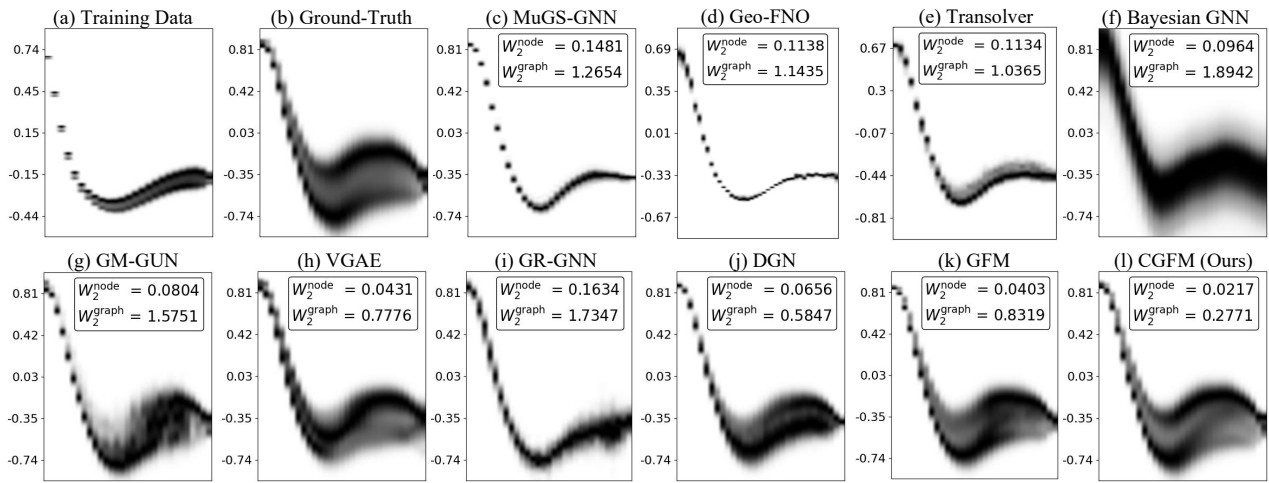

*Figure 6.* Probability density functions (PDFs) for a system on the Ellipse-Indist dataset. (a) PDF of incomplete training data comprising 10 consecutive states. (b) PDF of the full trajectory. (c)–(l) PDFs learned by each model trained on 10 time-steps per training trajectory. The deterministic and probabilistic baselines fail to recover the probability density of all the states of fully-developed flows. In contrast, CGFM produces a PDF that most closely matches the full trajectory and achieves the best performance in terms of node-wise and graph-wise measures.

*Table 8.* Mean coefficient of determination ($R^2$) of all models on the Ellipse datasets.

|  | -InDist | -Thin | -Thick | -LowRe | -HighRe |
|---|---|---|---|---|---|
| MuGS-GNN | $0.9858 \pm 0.0116$ | $0.9926 \pm 0.0026$ | $0.9721 \pm 0.0144$ | $\underline{0.9931 \pm 0.0052}$ | $0.9679 \pm 0.0179$ |
| Geo-FNO | $0.9807 \pm 0.0110$ | $0.9667 \pm 0.0195$ | $0.9309 \pm 0.0245$ | $0.9879 \pm 0.0058$ | $0.9624 \pm 0.0175$ |
| Transolver | $0.9802 \pm 0.0117$ | $0.9923 \pm 0.0029$ | $0.9609 \pm 0.0059$ | $0.9865 \pm 0.0073$ | $0.9631 \pm 0.0172$ |
| Bayesian GNN | $0.7008 \pm 0.2681$ | $0.4461 \pm 0.5981$ | $0.8073 \pm 0.1472$ | $0.6552 \pm 0.4136$ | $0.7401 \pm 0.3560$ |
| GM-GNN | $0.9357 \pm 0.0362$ | $0.9644 \pm 0.0135$ | $0.8776 \pm 0.0301$ | $0.9626 \pm 0.0292$ | $0.8940 \pm 0.0339$ |
| GR-GNN | $0.9442 \pm 0.0363$ | $0.9648 \pm 0.0170$ | $0.9091 \pm 0.0299$ | $0.9714 \pm 0.0178$ | $0.9073 \pm 0.0314$ |
| VGAE | $0.9762 \pm 0.0218$ | $0.9875 \pm 0.0110$ | $0.9351 \pm 0.0500$ | $0.9820 \pm 0.0182$ | $0.9609 \pm 0.0330$ |
| DGN | $\underline{0.9874 \pm 0.0067}$ | $0.9896 \pm 0.0052$ | $\underline{0.9795 \pm 0.0082}$ | $\underline{0.9931 \pm 0.0032}$ | $\underline{0.9765 \pm 0.0127}$ |
| GFM | $0.9809 \pm 0.0182$ | $\underline{0.9947 \pm 0.0040}$ | $0.9350 \pm 0.0486$ | $0.9858 \pm 0.0183$ | $0.9616 \pm 0.0309$ |
| CGFM (Ours) | $\mathbf{0.9950 \pm 0.0048}$ | $\mathbf{0.9959 \pm 0.0026}$ | $\mathbf{0.9893 \pm 0.0079}$ | $\mathbf{0.9968 \pm 0.0024}$ | $\mathbf{0.9846 \pm 0.0230}$ |

Here, $\mathbf{s}_{i,t}^{\text{gt}}$ denotes the ground-truth data, $\mathbf{s}_{i,t}$ denotes the predicted data, and $\bar{\mathbf{s}}_t^{\text{gt}}$ denotes the spatial mean of the ground-truth data across all nodes at $t$. When $R^2$ equals one, the predicted values exactly match the ground truth.

To quantify the distributional accuracy, we use the Wasserstein-2 distance by considering the predicted distribution as follows: (i) *1D distribution at each node* and (ii) *$|V|$-D distribution across the whole graph*. The node-wise distance ($W_2^{\text{node}}$) is calculated by computing the distribution at each node and averaging the results across all the nodes, it quantifies how well the model recovers local statistics such as the mean and variance at individual nodes. The graph-wise distance ($W_2^{\text{graph}}$) evaluates the discrepancy between the joint distributions across all the nodes in the graph, which reflects the ability of model to capture spatial correlations. Lower values indicate better distributional accuracy.

## F. Supplementary Results

### F.1. Performance Evaluation

**Learning the full-state distribution from incomplete trajectories.** We validate the result on the Ellipse and EllipseFlow tasks, where each training trajectory is intentionally restricted to $T_{\text{train}} = 10$ consecutive states from the 101-step fully developed simulations. Such trajectories cover only a limited segment of the vortex-shedding cycle and are therefore insufficient, by themselves, to reliably estimate the full statistics of the underlying periodic dynamics. To assess how well the models capture the full distribution of converged states, we construct the target distribution using 60 to 100 consecutive converged states, depending on the period length. The learned distribution is approximated by drawing 200 samples from

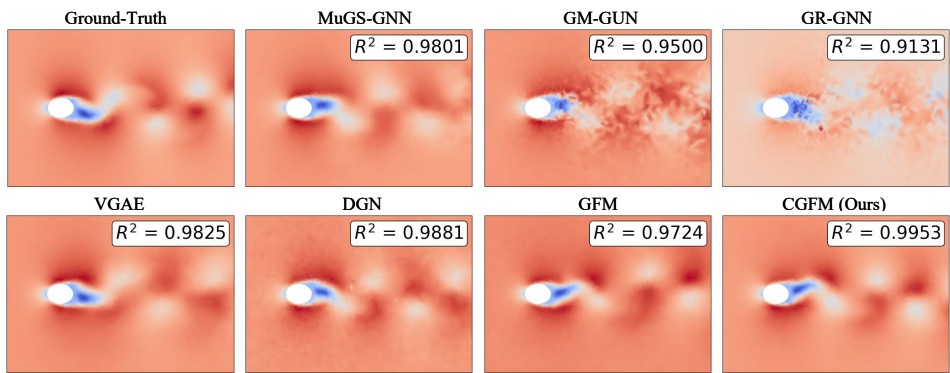

*Figure 7.* Samples of the velocity field around an elliptical cylinder generated by different models on the EllipseFlow-Indist dataset.

*Table 9.* Mean coefficient of determination ($R^2$) of all the models on the EllipseFlow datasets.

|  | -InDist | -Thin | -Thick | -LowRe | -HighRe |
|---|---|---|---|---|---|
| MuGS-GNN | 0.9778 ± 0.0165 | 0.9839 ± 0.0057 | 0.9646 ± 0.0224 | 0.9789 ± 0.0153 | 0.9679 ± 0.0181 |
| Bayesian GNN | 0.6970 ± 0.2086 | 0.6887 ± 0.2308 | 0.6707 ± 0.1906 | 0.6886 ± 0.2177 | 0.6949 ± 0.2085 |
| GM-GUN | 0.9577 ± 0.0187 | 0.9681 ± 0.0060 | 0.9244 ± 0.0295 | 0.9571 ± 0.0142 | 0.9452 ± 0.0189 |
| GR-GNN | 0.9434 ± 0.0259 | 0.9591 ± 0.0075 | 0.9156 ± 0.0241 | 0.9461 ± 0.0179 | 0.9357 ± 0.0240 |
| VGAE | 0.9768 ± 0.0129 | 0.9847 ± 0.0057 | 0.9577 ± 0.0257 | 0.9767 ± 0.0161 | 0.9678 ± 0.0176 |
| DGN | 0.9803 ± 0.0136 | 0.9797 ± 0.0096 | 0.9721 ± 0.0187 | 0.9794 ± 0.0134 | 0.9749 ± 0.0132 |
| GFM | 0.9625 ± 0.0226 | 0.9640 ± 0.0137 | 0.9311 ± 0.0271 | 0.9694 ± 0.0177 | 0.9551 ± 0.0252 |
| CGFM (Ours) | **0.9844 ± 0.0159** | **0.9881 ± 0.0096** | **0.9863 ± 0.0087** | **0.9871 ± 0.0082** | **0.9809 ± 0.0131** |

*Table 10.* Mean node-wise ($W_2^{\mathrm{node}}$) and graph-wise ($W_2^{\mathrm{graph}}$) Wasserstein-2 distances of all the models on the EllipseFlow datasets.

|  | $W_2^{\mathrm{node}} \downarrow$ | | | | | $W_2^{\mathrm{graph}} \downarrow$ | | | | |
|---|---|---|---|---|---|---|---|---|---|---|
|  | -InDist | -Thin | -Thick | -LowRe | -HighRe | -InDist | -Thin | -Thick | -LowRe | -HighRe |
| MuGS-GNN | 0.0384 | 0.0234 | 0.0662 | 0.0375 | 0.0455 | 6.0653 | 4.055 | 9.6839 | 6.0595 | 6.7768 |
| GM-GUN | 0.0290 | 0.0169 | 0.0475 | 0.0303 | 0.0333 | 7.2323 | 5.0208 | 10.738 | 7.1786 | 7.8532 |
| GR-GNN | 0.0314 | 0.0223 | 0.0488 | 0.0334 | 0.0353 | 7.5189 | 5.4245 | 11.0586 | 7.4956 | 8.1416 |
| VGAE | 0.0382 | 0.0228 | 0.0654 | 0.0367 | 0.0451 | 6.0452 | 3.9956 | 9.6393 | 5.9344 | 6.7953 |
| DGN | 0.0262 | 0.0180 | 0.0474 | 0.0261 | 0.0298 | 4.8395 | 3.3976 | 8.1250 | 4.7011 | 5.5331 |
| GFM | 0.0189 | 0.0176 | 0.0326 | 0.0186 | 0.0222 | 4.6029 | 4.2989 | 7.0181 | 4.4392 | 4.9133 |
| CGFM (Ours) | **0.0149** | **0.0101** | **0.0285** | **0.0166** | **0.0194** | **3.3410** | **2.7637** | **4.8366** | **3.4407** | **4.0169** |

CGFM and the baseline models under the same conditions. Distributional accuracy is measured using the node-wise and graph-wise Wasserstein-2 distances, which are denoted by $W_2^{\mathrm{node}}$ and $W_2^{\mathrm{graph}}$, respectively. We use these metrics to evaluate the model's performance on both in-distribution and out-of-distribution (OOD) test datasets, where the OOD cases involve Reynolds numbers and geometric parameters outside the training range.

Figure 6 shows the probability density function (PDF) for a system from the Ellipse-Indist dataset. We can see that the deterministic model fails to recover the PDF of the ground truth, while the generative models approximate it more faithfully, with CGFM yielding a PDF that most closely matches the ground truth. Tables 1 and 10 summarize the node-wise and graph-wise Wasserstein-2 distances, $W_2^{\mathrm{node}}$ and $W_2^{\mathrm{graph}}$, for the Ellipse and EllipseFlow tasks, respectively. Although MuGS-GNN achieves competitive $R^2$ values, its Wasserstein-2 distances remain significantly greater. This result reflects that deterministic models fit individual snapshots well but fail to recover the underlying probability of flow states. In contrast, CGFM consistently achieves the smallest Wasserstein-2 distances among all the baselines and maintains the highest $R^2$ values across all the datasets. These results indicate that CGFM is capable of effectively extracting the shared patterns across multiple conditions while accurately learning the full equilibrium-state distribution from incomplete trajectories.

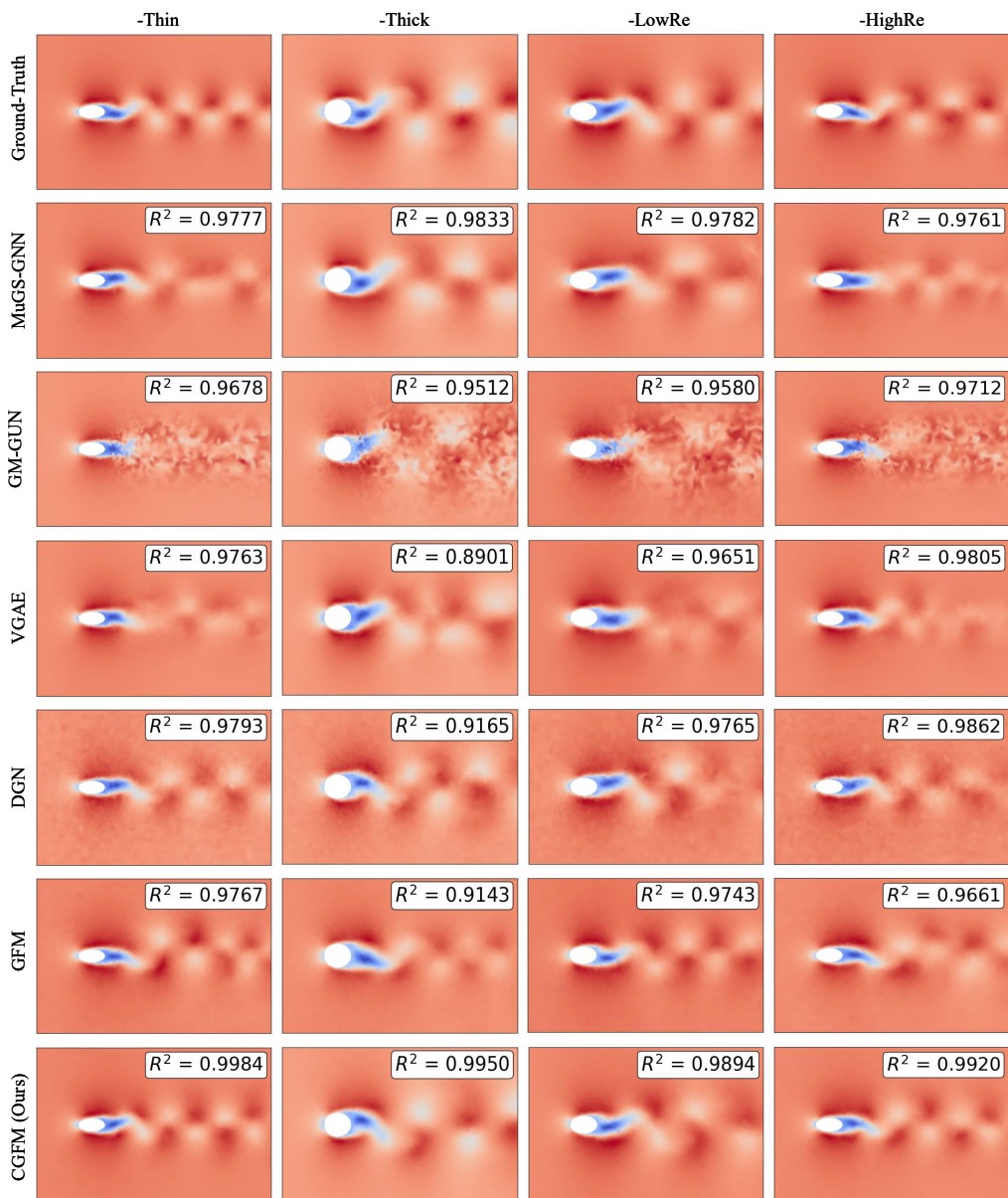

*Figure 8.* Comparison of generated samples from the OOD test case on the EllipseFlow dataset.

**Flow statistics derived from learned distributions.** A key objective of this work is to learn the full distribution of converged flow states from incomplete simulations, which allows physically relevant statistics to be efficiently computed by drawing multiple samples from the learned distribution. We therefore evaluate how well different models derive the turbulent kinetic energy (TKE) from their learned flow-state distribution on the EllipseFlow datasets. Table 11 summarizes the mean coefficient of determination of the TKE obtained by different models trained on only 10 consecutive flow states. Across all the datasets, our model achieves the highest $R^2_{\text{TKE}}$ values, whereas the baselines exhibit a noticeable degradation in TKE estimation under out-of-distribution settings, particularly in the Thick and HighRe cases. These results indicate that accurately learning the full equilibrium-state distribution allows CGFM to derive turbulence statistics such as TKE much more robustly than competing models do, even when evaluated under conditions unseen during training. This advantage is further illustrated in Figure 9, which shows that our model most closely matches the reference TKE fields on both in-distribution and out-of-distribution EllipseFlow test datasets.

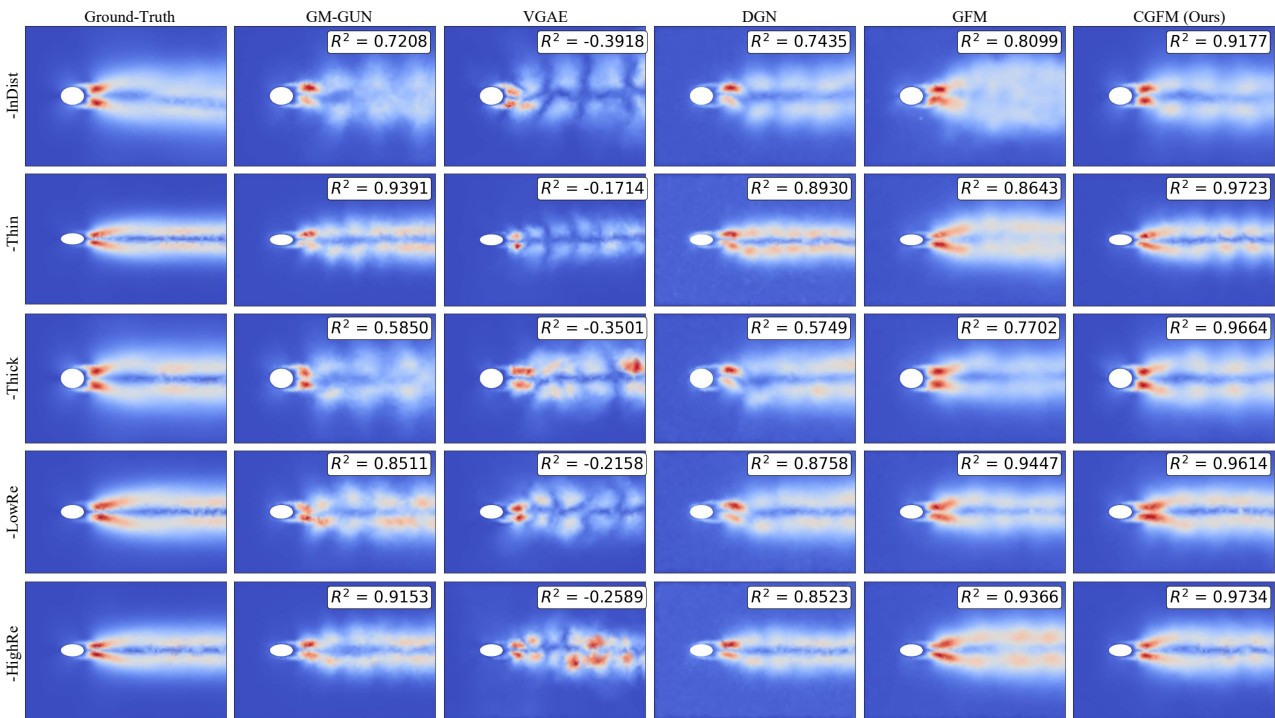

*Figure 9.* Turbulent kinetic energy computed from the learned flow-state distribution generated by the GM-GUN, VGAE, DGN, GFM, and CGFM for in-distribution and out-of-distribution test cases from the EllipseFlow dataset.

*Table 11.* Mean coefficient of determination of the TKE ($R^2_{\text{TKE}}$) on the EllipseFlow datasets.

|  | -InDist | -Thin | -Thick | -LowRe | -HighRe |
|---|---|---|---|---|---|
| GM-GUN | 0.8287 | 0.9411 | 0.6582 | 0.8408 | 0.7622 |
| VGAE | -0.3083 | -0.1727 | -0.438 | -0.2495 | -0.4122 |
| DGN | 0.8196 | 0.8838 | 0.6057 | 0.8603 | 0.7622 |
| GFM | 0.9089 | 0.8077 | 0.7344 | 0.9177 | 0.8694 |
| CGFM(Ours) | **0.9667** | **0.964** | **0.9224** | **0.9535** | **0.9306** |

*Table 12.* Efficiency comparison for generating 3,000 samples on the Wing task, including speed-up over the traditional numerical solver.

|  | CPU Inference time | CPU speed-up | GPU Inference time | GPU speed-up |
|---|---|---|---|---|
| DGN | 338.665 | 8.79× | 6.89 | 431.93× |
| GFM | 447.077 | 6.66× | 8.80 | 338.18× |
| **CGFM (Ours)** | **185.372** | **16.05×** | **4.91** | **606.11×** |

**Generalization to large-scale irregular 3D meshes.** To assess the ability of CGFM to generalize to large, irregular geometries, we evaluate it on the Wing task, which is a 3D turbulent-flow benchmark whose surface meshes contain approximately 7,000 nodes per sample. This benchmark provides a demanding test of both the scalability of our hierarchical graph architecture and its robustness to nonuniform spatial resolution. In this experiment, the target distribution over wing-surface pressure is represented by 2,500 consecutive flow states from high-fidelity simulations, whereas the predicted distribution is evaluated using 3,000 samples drawn from the learned model under the same geometric and flow conditions. As illustrated in Figures 5 and 10, the pressure fields generated by our model are visually closest to the ground truth, particularly in regions with strong three-dimensional effects such as the leading edge and wing tip, whereas the baseline models tend to either oversmooth or distort regions with strong pressure gradients. A quantitative comparison in terms of the coefficient of determination ($R^2$), node-wise and graph-wise Wasserstein-2 distances ($W_2^{\text{node}}$ and $W_2^{\text{graph}}$, respectively) is summarized in Table 2. Across all the settings, CGFM achieves the best overall balance of sample fidelity ($R^2$), node-wise distribution accuracy ($W_2^{\text{node}}$) and spatially coherent distribution accuracy ($W_2^{\text{graph}}$) among all the approaches. This finding demonstrates that CGFM not only accurately predicts individual turbulent realizations, but also captures the underlying

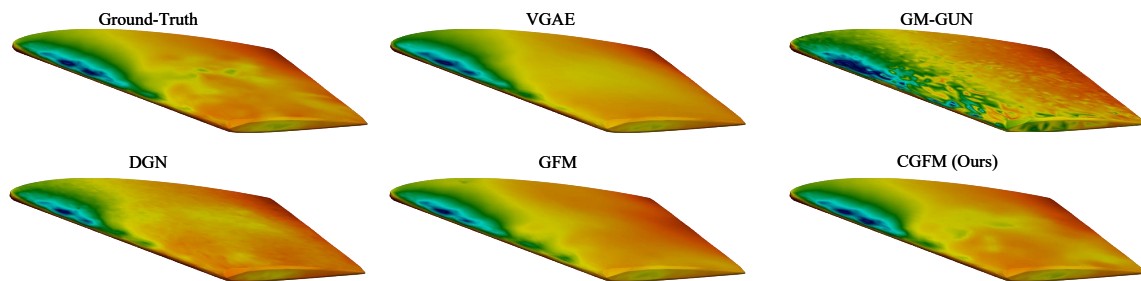

*Figure 10.* Pressure fields ($p$) generated by VGAE, GM-GUN, DGN, GFM, CGFM for a simulation on the Wing test dataset.

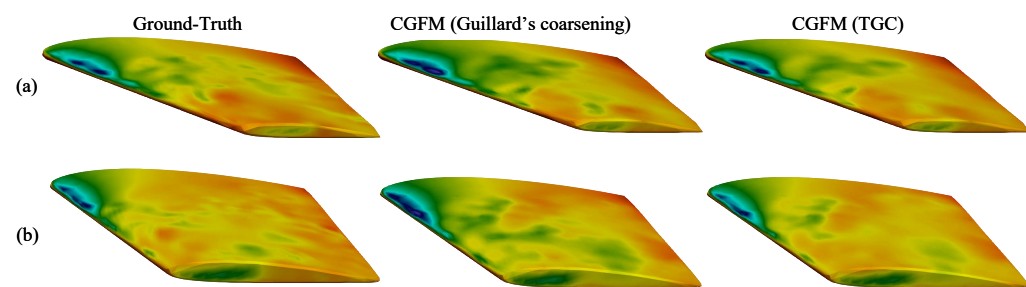

*Figure 11.* Comparison of the predicted surface pressure fields on the Wing test dataset. For each test system, we show the ground truth, CGFM with Guillard's coarsening, and CGFM with the proposed TGC scheme, where (a) and (b) correspond to two different test systems.

probability distribution of system states on large, irregular 3D meshes.

### F.2. Computational Speed-up over Numerical Solvers

To evaluate the practical efficiency of our method, we compare it against the traditional numerical solver. Although direct comparisons between different implementations are inherently difficult, we provide a practical estimate under the same CPU setting on the Wing task. Using OpenFOAM's PISO solver with 8 CPU threads, the simulation requires 2,976 minutes to evolve through the initial transient regime and then generate 2,500 equilibrium states, which is sufficient for computing aggregate statistics. In contrast, the generative models directly sample from the learned equilibrium distribution and therefore avoid the costly transient simulation stage. As reported in Table 12, CGFM generates 3,000 samples in 185.372 minutes on CPU, thus yielding a speed-up of $16.05\times$ over the numerical solver. Under the same CPU setting, the DGN and GFM achieve speeds-up of $8.79\times$ and $6.66\times$ speed-up, respectively. A further practical advantage of CGFM is that it can be executed efficiently on GPUs, whereas official OpenFOAM distributions lack GPU support. In our experiments, CGFM generates 3,000 samples in only 4.91 minutes on a single GPU, which corresponds to a speed-up of $606.11\times$ relative to the CPU runtime of the numerical solver. This result further highlights the deployment advantage of CGFM when fast repeated sampling is required. Overall, these results demonstrate the practical value of CGFM in settings where repeated statistical estimation with numerical solvers is computationally prohibitive.

### F.3. Ablation Studies

**Effects of the number of states per training trajectory.** A notable strength of our model is its ability to learn the full distribution of all plausible states from incomplete trajectories. In the Ellipse task, each training trajectory provides only 10 consecutive states, whereas full periods of the trajectories span 21 to 38 states. Despite this challenge, we found that our model exhibits only a slight degradation in distributional accuracy, and its performance with incomplete trajectories remains close to that obtained from fully observed trajectories. Figure 4 shows the distributional error of our model and the baselines on the Ellipse-Indist dataset. With respect to the node-wise distributional error, the $W_2^{\text{node}}$ values of the deterministic and probabilistic models decrease sharply as the number of training states increases, thereby indicating their sensitivity to trajectory length. In contrast, generative models remain largely unaffected. With respect to the graph-wise distribution error, the $W_2^{\text{graph}}$ values of the deterministic model also decrease significantly with more training states, while those of the

*Table 13.* Comparison of the mean coefficient of determination ($R^2$) for CGFM and its variant without hierarchy on the Ellipse and EllipseFlow datasets.

| Ellipse | -InDist | -Thin | -Thick | -LowRe | -HighRe |
|---|---|---|---|---|---|
| CGFM | 0.9950 ± 0.0048 | 0.9959 ± 0.0026 | 0.9893 ± 0.0079 | 0.9968 ± 0.0024 | 0.9846 ± 0.0230 |
| CGFM (w/o hierarchy) | 0.9793 ± 0.0187 | 0.9936 ± 0.0051 | 0.9359 ± 0.0482 | 0.9848 ± 0.0184 | 0.9598 ± 0.0310 |
| EllipseFlow | | | | | |
| CGFM | 0.9844 ± 0.0159 | 0.9881 ± 0.0096 | 0.9863 ± 0.0087 | 0.9871 ± 0.0082 | 0.9809 ± 0.0131 |
| CGFM (w/o hierarchy) | -10.9307 ± 0.5398 | -11.7975 ± 0.1160 | -9.6757 ± 0.1845 | -11.0709 ± 0.4704 | -10.6265 ± 0.5187 |

*Table 14.* Comparison of the mean node-wise ($W_2^{\text{node}}$) and graph-wise ($W_2^{\text{graph}}$) Wasserstein-2 distances for CGFM and its variant without hierarchy on the Ellipse and EllipseFlow datasets.

| | $W_2^{\text{node}}\downarrow$ | | | | | $W_2^{\text{graph}}\downarrow$ | | | | |
|---|---|---|---|---|---|---|---|---|---|---|
| Ellipse | -InDist | -Thin | -Thick | -LowRe | -HighRe | -InDist | -Thin | -Thick | -LowRe | -HighRe |
| CGFM | 0.0239 | 0.0164 | 0.0602 | 0.0166 | 0.0372 | 0.2734 | 0.1574 | 0.6592 | 0.1944 | 0.4415 |
| CGFM (w/o hierarchy) | 0.0328 | 0.0200 | 0.0744 | 0.0243 | 0.0464 | 0.6001 | 0.2477 | 1.3562 | 0.4588 | 0.7837 |
| EllipseFlow | | | | | | | | | | |
| CGFM | 0.0149 | 0.0101 | 0.0285 | 0.0166 | 0.0194 | 3.3410 | 2.7637 | 4.8366 | 3.4407 | 4.0169 |
| CGFM (w/o hierarchy) | 0.8032 | 0.8153 | 0.7837 | 0.8062 | 0.7989 | 81.8393 | 81.2748 | 81.4707 | 82.8217 | 81.7084 |

other models are stable overall. Notably, GM-GUN achieves the lowest $W_2^{\text{node}}$ among all the models when trained on long trajectories. However, its $W_2^{\text{graph}}$ values remain consistently high because it does not explicitly capture spatial correlations. Compared with the baseline models, our model exhibits the least sensitivity to the number of states per training trajectory, which demonstrates its ability to accurately learn the complete distribution of possible states from relatively short training trajectories. This ability makes it highly promising for practical engineering and scientific computing tasks.

**Hierarchical vs. Single-scale architecture in CGFM.** A key characteristic of the HieraGraphNet architecture of our CGFM model is its ability to perform message passing across multiresolution graph hierarchies through the Inter-MP module. To verify its effectiveness, we construct a single-scale variant of CGFM in which all the inter-level message passing has been removed and that operates only on the finest graph $\mathcal{G}_0$. To ensure a fair comparison, we use the same hyperparameter settings as in our CGFM model. In Tables 4 and 13–14, the $R^2$, $W_2^{\text{node}}$, and $W_2^{\text{graph}}$ metrics of our CGFM model and its variants are compared on the Ellipse, EllipseFlow and Wing datasets. In small-scale systems (e.g., the Ellipse dataset with approximately 70 nodes per graph), compared with its multi-scale counterpart, the single-scale CGFM model has a lower sample and distributional accuracy, while its performance remains sufficiently reliable for tasks of this scale. However, this behavior does not generalize to larger systems. The single-scale model shows markedly worse performance across all the metrics on both the EllipseFlow task (approximately 2.3k nodes per graph) and the Wing task (approximately 6.8k nodes per graph). Figure 12 further illustrates that the single-scale model fails to generate physically plausible flow states in the large-scale domain. These results indicate that message passing restricted to the original high-resolution graph, with its limited receptive field, fails to capture the inherent multi-scale dynamics and long-range dependencies present in fluid flows. In contrast, our CGFM model leverages a hierarchical graph structure to transfer inter-level information across graph hierarchies, which allows it to capture global physical patterns and multi-scale flow structures more effectively.

**TGC scheme vs. Guillard's coarsening** We evaluate the contribution of the proposed topology- and geometry-aware graph coarsening (TGC) scheme by replacing it with Guillard's coarsening and comparing the resulting performance on the Wing task. As shown in Table 4, substituting TGC scheme with the alternative coarsening strategy leads to a consistent decrease in both the sample accuracy and distributional accuracy. In particular, the CGFM model w/ Guillard variant exhibits higher $W_2^{\text{node}}$ and $W_2^{\text{graph}}$, which indicates that it fails to preserve the multi-scale spatial correlations required for modeling turbulent pressure distributions. A key reason for this performance gap lies in the geometric complexity of the Wing meshes, where node spacing varies substantially across the surface because of changes in curvature, taper, and twist. Our TGC scheme incorporates the local geometric density into the coarsening criterion, which ensures that nodes in geometrically dense or structurally intricate regions are preferentially retained while the global topological structure is preserved. This

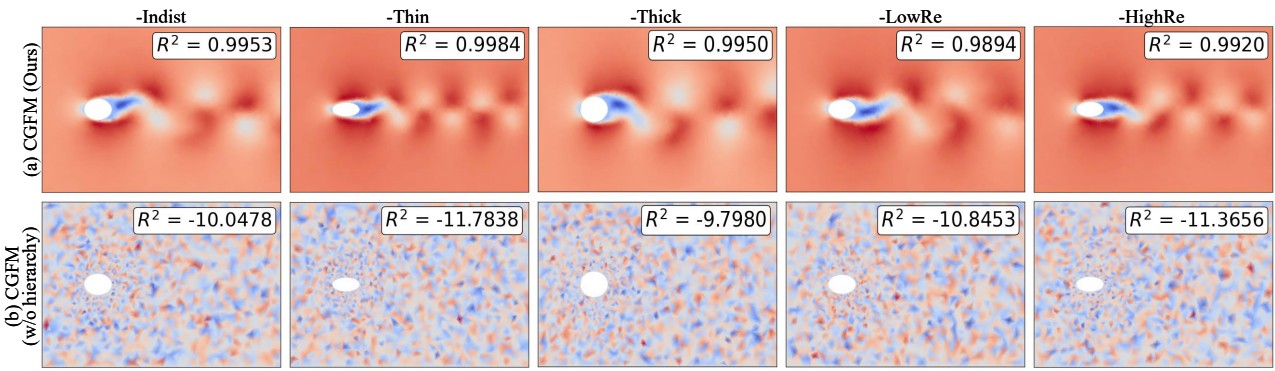

*Figure 12.* Samples generated by (a) CGFM and (b) its single-scale variant from the EllipseFlow dataset. The variant of CGFM fails to generate physically plausible flow states on the EllipseFlow dataset, because it cannot capture the multi-scale structures and long-range dependencies of the flow.

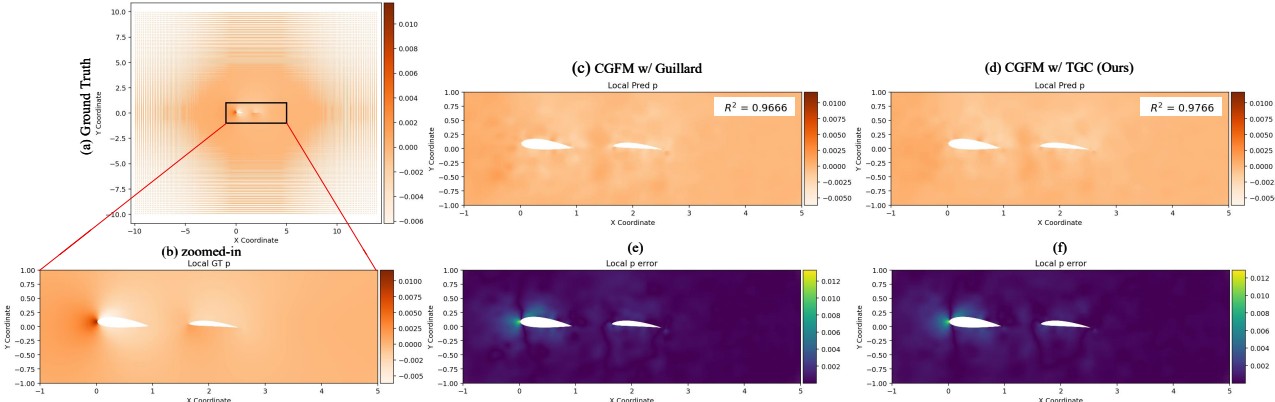

*Figure 13.* Visual comparison of different coarsening schemes. (a) Ground-truth field. (b) Zoomed-in view of the region marked by the black box in (a). (c) Result generated by replacing the TGC scheme with Guillard's coarsening algorithm. (d) Result of CGFM with TGC scheme (ours). (e) and (f) show the corresponding absolute error maps for (c) and (d), respectively.

joint topology–geometry scoring produces coarse graphs that better preserve critical flow structures and spatial correlations on complex wing surfaces. Visual comparisons in Figure 11 further illustrate these advantages. By maintaining a more faithful relative node distribution across scales, TGC scheme provides more informative multi-scale message passing within HieraGraphNet, ultimately achieving significantly improved distributional fidelity on large-scale 3D turbulent flows.

To evaluate the robustness of our method under genuine topology changes, we further test it on the CruiseAOA=5° subset of TandemFoilSet (Lim et al., 2026), a public computational fluid dynamics benchmark specifically designed for tandem-airfoil flow prediction. TandemFoilSet is intended to support learning-based prediction in multi-body aerodynamic configurations. The CruiseAOA=5° subset used in our experiments contains 784 tandem-airfoil cases, which provide a more challenging setting than single-body configurations because the underlying geometry becomes structurally more complex. We use 705 cases for training and 79 cases for testing. Each case contains, on average, 351,315 cells, and the dataset is simulated at Re = 500 and AoA = 5°. We compare CGFM w/ TGC (ours) against CGFM w/ Guillard, a variant that replaces TGC with Guillard's coarsening algorithm while keeping all other components unchanged. For each test case, we generate 50 samples and quantitatively evaluate the results in terms of $R^2$, $W_2^{\text{node}}$, and $W_2^{\text{graph}}$. As shown in Figure 13 and Table 15, CGFM w/ TGC consistently outperforms CGFM w/ Guillard. It improves $R^2$ by approximately 1.16% and reduces $W_2^{\text{node}}$ and $W_2^{\text{graph}}$ by approximately 18.5% and 17.0%, respectively. These results indicate that TGC scheme is more robust than the Guillard-based alternative in this more challenging multi-body setting.

**Effect of the number of hierarchy levels in HieraGraphNet.** The number of hierarchy levels in HieraGraphNet is a key factor in CGFM, as it determines the ability of the model to capture dynamics across multiple spatial resolutions and facilitate

*Table 15.* Comparison of CGFM and its variants in terms of $R^2$, $W_2^{\text{node}}$, and $W_2^{\text{graph}}$ on the TandemFoilSet dataset.

|  | $R^2 \uparrow$ | $W_2^{\text{node}} \downarrow$ | $W_2^{\text{graph}} \downarrow$ |
| --- | --- | --- | --- |
| CGFM w/ Guillard | 0.9637±0.0179 | 0.0054 | 12.744 |
| CGFM w/ TGC (Ours) | 0.9749±0.0151 | 0.0044 | 10.578 |

*Table 16.* Effects of the number of hierarchy levels on the accuracy and inference cost.

| Dataset | # hierarchy levels | $R^2$ | $W_2^{\text{node}}$ | $W_2^{\text{graph}}$ | Inference time (ms/sample) |
| --- | --- | --- | --- | --- | --- |
| *Ellipse* | 2 | 0.9725 ± 0.0262 | 0.0331 | 0.6415 | 26.216 |
|  | 3 | 0.9896 ± 0.0130 | 0.0289 | 0.4114 | 42.474 |
|  | 4 | **0.9950 ± 0.0048** | **0.0239** | **0.2734** | 42.573 |
| *EllipseFlow* | 2 | 0.9534 ± 0.0188 | 0.0227 | 5.1318 | 105.697 |
|  | 3 | 0.9623 ± 0.0264 | 0.0187 | 4.5393 | 130.227 |
|  | 4 | 0.9739 ± 0.0196 | 0.0173 | 3.8879 | 180.404 |
|  | 5 | **0.9844 ± 0.0159** | **0.0149** | **3.3410** | 230.202 |

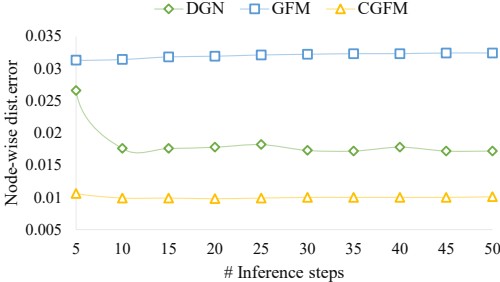

*Figure 14.* Node-wise distributional error versus inference steps on the Wing dataset.

long-range physical interactions on irregular meshes. To evaluate the effect of hierarchical depth, we compare CGFM variants with different numbers of hierarchy levels under the same experimental setup. As shown in Table 16, performance consistently improves as the number of hierarchy levels increases. On the Ellipse task, our 4-level model outperforms the 2-level variant in terms of $R^2$ value by approximately 2.3% while reducing $W_2^{\text{node}}$ and $W_2^{\text{graph}}$ by approximately 27.8% and 57.4%, respectively. The benefits become more pronounced on the larger and more complex EllipseFlow dataset. Our 5-level model outperforms the 2-level variant in terms of $R^2$ by approximately 3.3% and reduces $W_2^{\text{node}}$ and $W_2^{\text{graph}}$ by approximately 34.4% and 34.9%, respectively. These improvements confirm that deeper hierarchies strengthen the ability of the model to capture multi-scale dynamics and dependencies on fluid systems. Although the inference time increases with the number of hierarchy levels, the gains indicate that a deeper hierarchy offers a more favorable trade-off for challenging large-scale tasks.

**Effects of the number of inference steps.** The number of inference steps directly affects the trade-off between inference efficiency and distributional accuracy. To evaluate this sensitivity, we conducted an ablation study on the Wing dataset by varying the number of inference steps and measuring the resulting $W_2^{\text{node}}$ based on 3,000 generated samples. As shown in Figure 14, CGFM consistently achieves the best distributional recovery among all the compared methods and remains highly stable across different numbers of inference steps. In particular, CGFM achieves strong accuracy even with very few inference steps ($< 10$), which indicates that accurate sampling can already be achieved with only a small number of steps. In contrast, the DGN is more sensitive to the number of inference steps, and its error is noticeably higher when the number of steps is below 10. The results demonstrate that CGFM is substantially less sensitive to the number of inference steps and provides the most favorable balance between inference efficiency and distributional accuracy.

