# OpenReview forum: "Condition-Aware Graph Flow Matching for Modeling the Distributions of Complex Fluid Systems"
_ICML.cc/2026/Conference — ICML 2026 regular_

### Official Review · Reviewer_W47c · 2026-03-02

**Soundness:** 4
**Presentation:** 3
**Significance:** 3
**Originality:** 3
**Overall Recommendation:** 4
**Confidence:** 4

**Summary:**

This paper introduces Condition-Aware Graph Flow Matching (CGFM), a novel approach designed to model the full distribution of converged flow states for irregular physical systems using incomplete training data. The paper combines flow matching with a hierarchical graph structure (HieraGraphNet) and topology- and geometry-aware graph coarsening to generate high-fidelity equilibrium-state distributions. By training the model on short segments of equilibrium states, CGFM aims to efficiently sample diverse states and compute relevant statistical properties without relying on extensive numerical simulations. The experimental validation demonstrates the model's superiority over existing methods in both accuracy and generalization, with tests on both 2D and 3D fluid dynamics tasks.

**Compliance With Llm Reviewing Policy:**

Affirmed.

**Final Justification:**

I maintain my score as Weak Accept. The rebuttal addressed my concerns, and I have read the other authors' and reviewers' discussions. I think the authors addressed most of the reviewers' concerns (centering the scope on fluid dynamics, adding stronger baselines and related work, and specifying parameters). If this paper is accepted, the authors should incorporate the proposed experiments into the revision. Clarity can also be improved.

**Key Questions For Authors:**

1. Can CGFM be extended to other categories of physical systems, such as structural mechanics or electromagnetic simulations? If so, could the authors provide illustrative examples or preliminary experimental results?
2. How does performance change as the trajectory length becomes even shorter (e.g., fewer than 10 states)? Is there a practical lower bound on trajectory length below which performance degrades significantly?

**Limitations:**

Yes

**Strengths And Weaknesses:**

**Strengths**
1. The integration of flow matching with hierarchical graph learning and graph coarsening provides an effective framework for modeling complex physical systems, particularly in turbulent or chaotic regimes where trajectories are incomplete.
2. The theoretical foundation of CGFM is well-developed and convincingly presented.
3. The paper offers comprehensive experimental evaluations on multiple datasets, consistently showing that CGFM outperforms strong baseline methods.
4. The method exhibits strong generalization capability under out-of-distribution settings.

**Weaknesses**
1. The hierarchical message passing scheme and multi-resolution graph construction introduce additional computational overhead. Although the authors state that the coarsening step does not increase runtime due to preprocessing, a more detailed comparison of training time and memory consumption would further substantiate this claim.
2. While the paper highlights the ability to learn from short trajectories, it lacks a systematic analysis of the method’s sensitivity to the number of states per trajectory during training. Such an ablation study would strengthen the empirical evaluation.

---

> ### Author Rebuttal · Authors · 2026-03-31
>
> We sincerely thank the reviewer for the positive assessment of our work. Below, we address the reviewer’s questions and suggestions.
>
> 1.we have added a direct efficiency comparison on the Wing dataset using a single NVIDIA GeForce RTX 4090 GPU (24 GB). The results show that, despite introducing a hierarchical graph structure, CGFM is in fact more efficient in practice than the non-hierarchical GFM baseline on this large-scale 3D irregular-mesh task. In particular, compared with GFM, CGFM reduces GPU memory usage from 3.80 GB to 1.59 GB (approximately 58.2% reduction) and decreases the inference time for generating 3000 samples from 527.8 s to 294.39 s (approximately 44.2% faster), while also achieving stronger accuracy. These results support our claim that the hierarchical coarse-to-fine design improves scalability on large irregular systems rather than introducing prohibitive overhead.
> We will revise the manuscript to make this point more explicit and include the corresponding runtime and memory statistics in the experimental discussion. We have included an anonymous supplementary repository for the reviewer’s reference, which contains the corresponding experimental tables and figures: \texttt{[https://anonymous.4open.science/r/cgfm-F54D]}.
>
> 2. We appreciate this suggestion. In fact, the current manuscript already includes a systematic analysis of the effect of the number of states per training trajectory in the subsection: “Effects of the number of states per training trajectory.”  In this experiment, conducted on the Ellipse-InDist dataset, we compare CGFM with six baselines while varying the number of observed states in each training trajectory. As discussed in the paper and shown in Figure 3, our model remains largely robust even when trained on incomplete short trajectories. In the Ellipse task, each training trajectory provides only 10 consecutive states, while a full period typically spans 21 to 38 states. Despite this substantial truncation, CGFM shows only a slight degradation in distributional accuracy, and its performance with incomplete trajectories remains close to that obtained from fully observed trajectories.
> More specifically, the results show that deterministic and probabilistic baselines are much more sensitive to trajectory length: their W_2^nodevalues decrease sharply as more training states are provided. By contrast, the generative models (especially CGFM) are much less affected. Compared with the baselines, CGFM exhibits the least sensitivity to the number of states per training trajectory, which supports our main claim that the model can recover the full distribution of plausible states from relatively short trajectory segments.
> We agree, however, that this point could be made more visible in the manuscript. In the revised version, we will highlight this experiment more clearly in both the main text and the discussion, so that the connection between the “short-trajectory learning” claim and the supporting empirical evidence is more explicit.
>
> 3. We thank the reviewer for this important question. In principle, CGFM is not restricted to fluid dynamics. The framework only assumes that the physical state can be represented on a graph/mesh with associated conditioning variables, and that the target is to learn a conditional equilibrium-state distribution from incomplete trajectory data. Under this formulation, the method is applicable in principle to other mesh-based physical systems.  That said, we agree that the current experimental section is focused on fluid systems, and we do not yet provide empirical evidence on other physical domains. We will revise the manuscript to better qualify the scope of our claims and explicitly mention extension to other graph-based PDE systems as an important future direction.

---

> > ### Author Rebuttal · Reviewer_W47c · 2026-04-01
> >
> > I have read the rebuttal and other reviewers' comments. I will keep my rating and see if the authors' response adequately addresses the other reviewers' concerns.

---

> > > ### Author Response · Authors · 2026-04-02
> > >
> > > Thanks again for your time and consideration.

---

### Official Review · Reviewer_WEb3 · 2026-03-12

**Soundness:** 1
**Presentation:** 3
**Significance:** 1
**Originality:** 2
**Overall Recommendation:** 4
**Confidence:** 4

**Summary:**

This paper proposes a generative framework designed to model the full equilibrium-state distributions in complex, unsteady physical systems with particular adaptability to irregular sharp-gradient geometries

**Compliance With Llm Reviewing Policy:**

Affirmed.

**Final Justification:**

The authors have addressed my comments on comparison with baseline models and out of distribution performance.

**Key Questions For Authors:**

We would like to increase the score if the authors can clarify why in the numerical section, most popular physics learning model is missing for baselines.

**Limitations:**

Yes

**Strengths And Weaknesses:**

Strengths:

1)	The model demonstrates a remarkable ability to recover the full probability density function of converged states even when trained on very short segment
2)	All the information is well presented.

Weaknesses:

1)	The paper highlights that existing generative models often neglect the "intricate effects of conditioning variables". However, the added value relative to existing "Graph Flow Matching" (GFM) baselines could be more explicitly quantified
2)	While the paper claims "superior interpolation and extrapolation" , the methodology relies on a shared-structure assumption where the true system lies within the learned hypothesis space.
3)	My biggest concern is that this paper lacks comparison with several important baselines, such as geo-FNO, transolver, transolver++, etc. The lack of sufficient baseline makes it hard to evaluate the true advantage of the proposed model, compared with the state-of-the-art.

---

> ### Author Rebuttal · Authors · 2026-03-31
>
> 1. We thank the reviewer for this suggestion and agree that the added value of CGFM over the existing GFM baseline should be quantified more explicitly. To address this, we have now added a direct efficiency comparison on the Wing dataset, which is a particularly challenging setting involving large-scale 3D irregular meshes, multi-scale turbulent structures, and long-range spatial dependencies. Compared with GFM, CGFM is specifically designed for representing large-scale irregular systems. In particular, we introduce a topology- and geometry-aware graph coarsening scheme to construct a coarse-to-fine hierarchy of unstructured mesh graphs. Based on this hierarchy, HieraGraphNet performs cross-level message passing, which helps capture multi-scale dynamic properties and facilitates long-range interactions in physical systems. By contrast, GFM applies flow matching more directly on the original graph without explicit multi-scale processing. As a result, on large irregular data such as Wing, GFM requires higher memory usage and incurs slower inference.  CGFM achieves higher accuracy while also being more efficient. Compared with GFM, CGFM reduces GPU memory usage from 3.80 GB to 1.59 GB (a reduction of about 58.2%) and decreases inference time for generating 3000 samples from 527.8 s to 294.39 s (about 44.2% faster). We have included an anonymous supplementary repository for the reviewer’s reference, which contains the corresponding experimental tables and figures: \texttt{[https://anonymous.4open.science/r/cgfm-F54D]}.
>
> 2. While the paper claims "superior interpolation and extrapolation", the methodology relies on a shared-structure assumption where the true system lies within the learned hypothesis space.
> We thank the reviewer for this important comment. We agree that the wording around “superior interpolation and extrapolation” should be made more precise.  Our intention was not to claim unconstrained extrapolation beyond the underlying structure represented in the training data. Rather, our claim is that CGFM demonstrates stronger generalization under out-of-distribution physical settings within the considered task family, where shared physical structure exists across conditions. To evaluate this explicitly, we tested generalization to out-of-distribution Reynolds numbers and ellipse thicknesses on both the Ellipse and EllipseFlow tasks. These experiments include both in-distribution and out-of-distribution settings, allowing us to assess whether the learned model remains accurate when the physical parameters move beyond the training distribution.
> The results show that CGFM maintains strong performance on the OOD datasets, with both sample accuracy and distributional accuracy remaining comparable to those on the in-distribution datasets, as shown in Tables 7–8 and 11–12 of the paper. This suggests that the model is able to generalize effectively across unseen physical conditions within the target problem family, rather than only memorizing individual training trajectories.
>
> 3. We sincerely thank the reviewer for the positive assessment of our paper, especially for recognizing the model’s ability to recover the full equilibrium-state distribution from very short trajectory segments and the clarity of the presentation. Below, we address the reviewer’s concerns. We fully agree with the reviewer that comparisons with strong domain-specific baselines are important. In response, we have now added comparisons with Geo-FNO and Transolver on the Ellipse benchmark, including both in-distribution (InDist) and out-of-distribution (OOD) splits (Thin, Thick, LowRe, HighRe). CGFM consistently outperforms both Transolver and Geo-FNO in terms of both distributional accuracy and sample accuracy across all evaluated settings. On the in-distribution split, CGFM substantially reduces both node-wise and graph-wise Wasserstein distances relative to the two baselines. The gains remain similarly strong on the OOD splits, including the more challenging Thick and HighRe settings. In addition, while Transolver and Geo-FNO perform reasonably well on in-distribution data, their R^2drops more clearly under OOD evaluation, whereas CGFM maintains consistently high R^2, suggesting stronger generalization under unseen conditions.

---

> > ### Author Rebuttal · Reviewer_WEb3 · 2026-04-04
> >
> > I have changed my recommendation to week accept

---

> > > ### Author Response · Authors · 2026-04-04
> > >
> > > We sincerely thank you for raising the score and for your positive feedback.

---

### Official Review · Reviewer_xv52 · 2026-03-13

**Soundness:** 2
**Presentation:** 3
**Significance:** 3
**Originality:** 3
**Overall Recommendation:** 5
**Confidence:** 5

**Summary:**

The paper introduces CGFM (Condition-aware Graph Flow Matching), a generative framework for learning the equilibrium state distribution of complex physical systems from short trajectory data.  CGFM combines condition-aware flow matching with a hierarchical graph neural network (HieraGraphNet) that performs multi-level message passing over mesh-based graph structures.  A topology- and geometry-aware graph coarsening scheme is introduced to handle irregular geometries and adaptively allocate spatial resolution.  The method is evaluated on canonical fluid mechanics-based tasks.

**Compliance With Llm Reviewing Policy:**

Affirmed.

**Final Justification:**

The paper is a good piece of work and would like to keep my positive outlook for the paper. The paper should be accepted.

**Key Questions For Authors:**

Key questions/suggestions/clearifications are as follows

1. The theoretical analysis, while correct, has no empirical verification. The experimental scope is limited to one physical domain at a modest scale, yet the paper frames itself as addressing complex physical systems broadly.  How will it generalize to other domains?
2. Key practical details (compute costs, ODE steps) and ablations (hierarchy depth, temporal conditioning) are missing.
3. Why was MMFM excluded from the baselines, given that it is the most closely related prior work on flow matching across varying physical conditions?
4. Have you evaluated CGFM on any non-fluid-dynamics PDE system (e.g., heat equation, structural mechanics)?

**Limitations:**

Yes.

**Strengths And Weaknesses:**

**Strength**

1. The paper is well-organized and easy to follow. The problem formulation and the condition-aware graph flow-matching framework are solid and likely to be useful to the community.
2. Directly modelling equilibrium distributions rather than rolling out trajectories is an important contribution.
3.The theoretical analysis is correct and provides good intuition, although no empirical connection between the theory and experiments is provided
4. The experiments demonstrate clear empirical advantages for CGFM, particularly in distributional accuracy ($W_2$ metrics), large-scale generalisation to irregular 3D meshes and OOD splits.

**Weaknesses**

1. All datasets are from the same physical domain (fluid dynamics). No evidence is provided for other physical systems, such as heat transfer.
2. Several key design choices are not ablated: the number of hierarchy levels $L$, the number of ODE integration steps $K$ at inference, and the temporal conditioning mechanism. The TGC vs Guillard ablation is limited to one dataset (Wing) and yields modest improvements.
3. MMFM, which extends flow matching to complex dynamic systems across time and varying conditions, is cited in related work but not compared against.
4. Training time, inference time, memory usage, and number of ODE steps K are never reported or compared against baselines.

---

> ### Author Rebuttal · Authors · 2026-03-31
>
> We sincerely thank the reviewer for the careful reading and constructive suggestions.
> 1. We agree with the reviewer that the current empirical scope is centered on fluid dynamics, and that the manuscript should describe this scope more carefully. Our motivation for focusing on fluid systems is that they provide a demanding testbed where three difficulties arise simultaneously: (1) equilibrium distribution learning from incomplete short trajectories, (2) large irregular meshes, and (3) multi-scale spatial interactions. At the same time, the CGFM model itself is not restricted to fluid-specific operators. The method only assumes that the physical state is represented on a graph/mesh with associated conditions, and that the target is to learn a conditional equilibrium-state distribution from incomplete trajectory data. In this sense, the model is in principle applicable to other PDE systems on irregular domains. We will therefore revise the manuscript to better qualify our claims. Extending the benchmark suite to non-fluid systems is an important direction for future work.
> 2. We agree that hierarchy depth, ODE steps, and compute costs are important and should be reported explicitly.
> For hierarchy depth, we conducted an additional ablation on Ellipse and EllipseFlow. The results can be seen in \texttt{[https://anonymous.4open.science/r/cgfm-F54D]}. These results show a clear trade-off between accuracy and inference cost. As the number of hierarchy levels increases, performance consistently improves in terms of R^2, W_2^node, and W_2^graph, while inference time also increases. This confirms that the hierarchical design is not arbitrary: deeper hierarchies improve multi-scale representation and long-range interaction modeling, especially on the more challenging task. We will include this ablation in the revised manuscript and clarify how the chosen hierarchy depth balances accuracy and efficiency. We also agree that the number of ODE steps should be reported explicitly. In response, we conducted an additional sensitivity study on the Wing dataset by varying the number of ODE steps and evaluating w_2^nodecomputed from 3000 generated samples. The results show that CGFM can maintain high distributional accuracy with very few ODE steps (< 10). We will add this ablation and report the chosen K clearly in the revised manuscript. We further agree that practical efficiency should have been reported more explicitly. To address this, we added runtime and memory measurements on the Wing dataset (large-scale 3D meshes). Training time is reported as the wall-clock time for one epoch, where each epoch consists of 1000 iterations. For inference efficiency, we report the wall-clock time required to generate 3000 samples. These results show that CGFM achieves the best practical efficiency among the compared generative models on the Wing task, with the lowest GPU memory usage, lowest per-epoch training time in our implementation, and fastest inference for generating 3000 samples.
>
> 3. Why MMFM was not included as a baseline.
> We thank the reviewer for raising MMFM as a relevant prior work. MMFM is evaluated on low-dimensional synthetic time-course data and single-cell RNA-seq data, where each condition is observed at multiple time points and the main goal is to model or interpolate distributions across time and conditions. In contrast, CGFM is designed for mesh-based physical systems, where each sample is defined on a large irregular graph with hundreds to thousands of nodes, and the goal is to recover the equilibrium-state distribution from incomplete short trajectories under varying physical conditions. As a result, MMFM is not a plug-and-play baseline for our setting.
> To ensure a fair and domain-relevant evaluation of CGFM, we instead added comparisons with Transolver and Geo-FNO, which are both strong and widely recognized baselines for PDE learning on irregular geometries. In particular, Transolver is highly competitive for irregular-mesh PDE learning and long-range interaction modeling, while Geo-FNO is a representative and influential geometry-aware neural operator baseline. We believe these models provide a more rigorous and meaningful test of CGFM in our target domain than a direct comparison to MMFM.
> We have now added these comparisons on the Ellipse benchmark, including both in-distribution (InDist) and out-of-distribution (OOD) splits (Thin, Thick, LowRe, HighRe). The results demonstrate that CGFM consistently outperforms both Transolver and Geo-FNO in terms of distributional accuracy and sample accuracy. In terms of sample accuracy, Transolver and Geo-FNO achieve relatively good performance on the in-distribution split, whereas their accuracy drops on the OOD splits. By contrast, CGFM preserves high R^2across all evaluated settings, suggesting that it has stronger extrapolation ability to unseen conditions.

---

> > ### Author Rebuttal · Reviewer_xv52 · 2026-04-03
> >
> > I have read the rebuttal and other comments. I would like to  keep my positive rating.

---

> > > ### Author Response · Authors · 2026-04-04
> > >
> > > Thank you once again for your time and consideration.

---

### Official Review · Reviewer_GSrU · 2026-03-13

**Soundness:** 2
**Presentation:** 2
**Significance:** 2
**Originality:** 2
**Overall Recommendation:** 2
**Confidence:** 4

**Summary:**

The paper proposes Condition-aware Graph Flow Matching (CGFM), a generative framework designed to learn the equilibrium distributions of complex physical systems (such as fluid dynamics) from limited, incomplete short-trajectory data. The method combines Flow Matching (FM) with a hierarchical graph neural network (HieraGraphNet) to handle unstructured meshes and multi-scale dynamics. The approach is validated on 2D and 3D fluid simulation datasets, demonstrating its ability to perform condition-aware interpolation and distributional recovery.

**Compliance With Llm Reviewing Policy:**

Affirmed.

**Final Justification:**

The added results on the traditional numerical solver (OpenFOAM) comparison and the multi-body TandemFoilSet evaluation are helpful, and they partially address my concerns regarding inference efficiency and topological changes. However, I still concern about the novelty of this paper. And I believe the paper needs a clearer justification of its limitations in highly chaotic regimes in the main text. I would like to keep my score.

**Key Questions For Authors:**

- Latency Analysis: What is the average wall-clock time to generate a single 3D Wing sample? How does the speed-up compare to a traditional solver when the goal is to compute aggregate statistics?
- ODE Step Sensitivity: How does the number of ODE integration steps affect the Wasserstein-2 distance? Is it possible to maintain distributional accuracy with very few steps (e.g., < 10)?
- Scalability: If the boundary conditions involve a topological change (e.g., from a single wing to a bi-plane configuration), would the TGC scheme require significant manual retuning?
- Efficiency: What is the current inference complexity of the model compared to the baselines? Why not consider newer flow based approaches, such as MeanFlow and Shortcut, which are more efficient at inference?

**Limitations:**

yes

**Strengths And Weaknesses:**

Strengths:
This paper combines conditional flow matching and hierarchical graph representation learning to provide an efficient and original solution for recovering the equilibrium state distribution of complex physical systems from short trajectory data of unstructured grids.

Weaknesses:
- Inference Latency and Sampling Cost: While the paper emphasizes avoiding long-time numerical integration, Flow Matching requires solving an ODE at inference time. For large-scale 3D meshes (like the Wing task), this could still involve significant computational overhead. The paper lacks a detailed wall-clock time comparison between CGFM sampling and traditional solvers or autoregressive GNN roll-outs at equivalent accuracy levels.
- Generalizability of the TGC Scheme: The coarsening scheme (TGC) relies heavily on local geometric density. Its robustness is not fully explored for scenarios where the geometry might undergo topological changes or where the sharp gradients are highly non-stationary across different conditions.
- Experimental Breadth: While a 3D case is included, the focus is quite narrow within the fluid dynamics domain. The datasets used are also relatively simple. It is unclear if the method maintains its distributional accuracy in more chaotic or highly turbulent regimes.
- Novelty: This paper appears to be a simple combination of flow matching and hierarchical graphs, lacking originality.

---

> ### Author Rebuttal · Authors · 2026-03-31
>
> We appreciate the reviewer’s comments and suggestions, which will help enhance the quality and clarity of the paper. Our responses to the reviewer’s concerns are provided below.
> 1. Inference Latency and Sampling Cost. We have added explicit wall-clock measurements on the Wing dataset.
> All efficiency measurements were conducted on the Wing dataset using a single NVIDIA GeForce RTX 4090 GPU (24 GB). Running time is measured by the time to complete one epoch, which contains 1000 iterations. For inference efficiency, we report the wall-clock time required to generate 3000 samples for each generative model. To assess the practical efficiency of generative modeling on large-scale 3D meshes, we compared DGN, GFM, and CGFM on the Wing dataset in terms of model size, GPU memory usage, training time per epoch, and inference time for generating 3000 samples. The results can be seen in \texttt{[https://anonymous.4open.science/r/cgfm-F54D]}. These results show that CGFM is not only more accurate in distributional recovery, but also more efficient at inference than both DGN and GFM. In particular, CGFM reduces inference time by about 28.7% compared with DGN and 44.2% compared with GFM, while also using substantially less GPU memory. Its per-epoch runtime is also slightly lower than DGN in our implementation.
>
> 2.We agree that the number of ODE integration steps is a key practical factor. We therefore conducted an additional sensitivity study on the Wing dataset by varying the number of ODE steps and evaluating $w_2^{node}$ computed from 3000 generated samples. The results show that CGFM can maintain high distributional accuracy with very few ODE steps (< 10). This indicates that the learned conditional transport path is stable and does not require many integration steps to produce accurate samples.
>
> 3. Generalizability of the TGC Scheme.
> Our current work studies distribution learning within a mesh-based problem family, where geometry and physical conditions vary across samples. In this setting, the TGC hierarchy is constructed automatically from each input mesh based on local geometric density and connectivity, and does not require manual redesign across the datasets considered in this paper.
> Importantly, the Wing task already involves geometric and physical variability. The wing geometry varies across relative thickness, taper ratio, sweep angle, and twist angle, and these changes directly affect the flow behavior and the distribution of sharp pressure gradients. Moreover, this task is specifically designed to evaluate performance in a challenging regime: it involves 3D turbulent flow at Re∼2×10^6, where vortices spontaneously form and dissipate at different locations on the wing surface. Therefore, TGC is evaluated not only on nearly identical geometries, but on a family of 3D wing configurations with meaningfully different aerodynamic characteristics and complex turbulent flow structures. We agree that testing on even broader or more chaotic regimes would further strengthen the paper, and we will present this as a limitation and future direction.
>
> 4.Novelty.
> We respectfully disagree with the assessment that the method is merely a simple combination of flow matching and hierarchical graphs. The contribution of CGFM is problem-driven and targets a setting not addressed by standard flow-based models: learning equilibrium-state distributions from incomplete short trajectories on large irregular physical meshes. More specifically, the originality lies in the following aspects: (1)Our goal is not long-horizon trajectory rollout, but direct recovery of the full distribution of plausible equilibrium states from short incomplete trajectory segments under varying physical conditions.
> (2)A \textit{topology- and geometry-aware graph coarsening} scheme is introduced to generate a coarse-to-fine hierarchy of unstructured mesh graphs, which is critical for representing complex geometries with spatially localized high gradients,
> (3)With this hierarchy, we performs cross-level message passing to capture the multi-scale dynamic properties and facilitate the long-range interactions of physical contexts. Thus, the value of CGFM is not just combining two existing modules, but designing a unified framework that integrates conditioning variables (e.g., system mesh, boundary conditions, and physical parameters) into a flow-based generative model that is tailored to short-trajectory distribution learning on large irregular physical systems.
>
> 5. We additionally added comparisons with strong domain-specific SOTA PDE baselines, namely Transolver and Geo-FNO. These baselines are highly competitive in irregular-geometry scientific learning, and provide a more domain-relevant and rigorous evaluation of CGFM in our application setting.  We agree that MeanFlow and Shortcut are promising future directions. We will mention them explicitly in the revised manuscript as valuable future extensions for further improving inference efficiency.

---

> > ### Author Rebuttal · Reviewer_GSrU · 2026-04-04
> >
> > Thank you to the authors for the detailed rebuttal, the new ODE step sensitivity study, and the inclusion of the additional Transolver and Geo-FNO baselines. I appreciate these clarifications, but I still have some remaining concerns.
> >
> > A core issue presented by the paper is the prohibitive computational cost of using traditional numerical solvers to explore the full statistical distribution of complex physical systems. While the authors added wall-clock comparisons demonstrating that CGFM is faster than other generative models, they did not directly answer my question regarding how the speed-up compares to a traditional solver (e.g., OpenFOAM) when computing aggregate statistics. Without this baseline comparison, it is difficult to fully assess the practical utility and true efficiency gain of the proposed surrogate in a real-world pipeline.
> >
> > Furthermore, a central concept presented by this paper is the topology- and geometry-aware coarsening (TGC) scheme. In the rebuttal, the authors clarified that TGC handles geometric variations (such as changes in sweep, twist, or taper) automatically. However, my specific concern regarding true topological changes such as transitioning from a single wing to a bi-plane configuration remains empirically unaddressed. It is still unclear if the heuristic coarsening metric remains robust when the underlying mesh topology fundamentally changes.

---

> > > ### Author Response · Authors · 2026-04-08
> > >
> > > Response to question 1:
> > >
> > > We appreciate the reviewer’s suggestion. We have added an explicit comparison with the traditional numerical solver. For the Wing task, OpenFOAM’s PISO solver, running on 8 CPU threads, required 2,976 minutes to simulate the initial transient phase and then generate 2,500 equilibrium states, which was reported to be sufficient for computing aggregate statistics. This reflects the standard workflow of traditional solvers: they must first evolve the system through a long transient regime until statistical stationarity is reached, and only then can additional samples be collected to estimate aggregate statistics.
> > >
> > > In contrast, our generative models directly sample from the learned equilibrium distribution and therefore avoid the expensive transient simulation stage. A further practical advantage of CGFM is that, unlike the traditional solver (official OpenFOAM distributions lack GPU support), it can be executed efficiently on GPUs. As shown in Table 1 (https://anonymous.4open.science/r/CGFM_1-47CC/Tab1.png), CGFM generates 3,000 samples in 185.372 minutes on CPU and 4.91 minutes on GPU, corresponding to 16.05× and 606.11× speed-up over the numerical solver, respectively. For reference, DGN achieves 8.79× / 431.93× speed-up and GFM achieves 6.66× / 338.18× speed-up on CPU / GPU. These results support the practical value of CGFM in settings where repeated statistical estimation with numerical solvers is computationally prohibitive.
> > >
> > > Response to question 2:
> > >
> > > Thank you for your reply and additional comments. To address the reviewer’s concern about whether the TGC scheme remains effective under genuine topology changes, we further evaluate our method on the CruiseAOA=5° subset from TandemFoilSet, a public CFD benchmark specifically designed for tandem-airfoil flow prediction. TandemFoilSet is intended to support learning-based prediction in multi-body aerodynamic configurations. The CruiseAOA=5° subset used in our experiments contains 261 tandem-airfoil cases, which provide a more challenging setting than single-body configurations because the underlying geometry becomes structurally more complex. We use 234 cases for training and 27 cases for testing. Each case contains, on average, 351,315 cells, and the dataset is simulated at Re = 500 and AoA = 5°.
> > >
> > > We compare CGFM w/ Guillard and CGFM w/ TGC (ours) on this dataset by replacing the proposed TGC scheme with Guillard’s coarsening algorithm while keeping the rest of the framework unchanged. For each test case, we generate 50 samples and evaluate the results using three complementary metrics: sample fidelity measured by ($R^2$), node-level distribution accuracy measured by ($W_2^{\text{node}}$), and spatially coherent distribution accuracy measured by ($W_2^{\text{graph}}$). As shown in Fig. 1 and Table 2 (https://anonymous.4open.science/r/CGFM_1-47CC), CGFM w/ TGC consistently outperforms CGFM w/ Guillard, yielding approximately 0.7% improvement in ($R^2$), along with 13.4% and 7.1% reductions in ($W_2^{\text{node}}$) and ($W_2^{\text{graph}}$), respectively. These results indicate that TGC is more robust than the Guillard-based alternative in this more challenging multi-body setting.

---

### Decision · Program_Chairs · 2026-04-30

**Decision:**

Accept (regular)

**Comment:**

The paper proposes a condition-aware graph glow matching method that combines conditional flow matching with a hierarchical GNN (HieraGraphNet) to model equilibrium distributions from short trajectory data. The reviewers have highlighted the importance of the problem. Besides, the integration of flow matching with hierarchical graph representations and the proposed coarsening scheme are well-motivated, and the paper is clearly written. Empirical results show strong performance on fluid dynamics tasks.

Nevertheless, the reviewers have identified several limitations of the paper. First, the experimental scope is narrow, being limited to fluid dynamics. Besides, the reviewers noted that relevant baselines were missing. During the response period, the authors provided additional experiments which addressed this concern (note: this was partially done through anonymous repositories which were inactive by the time I was preparing this metareview). Reviewer GSrU's remaining concerns about the comparison to a traditional solver (OpenFOAM) and the robustness of the heuristic metric to changes in the mesh topology appear to have been addressed (again via links to external, anonymous repositories that included additional tables/figures).